# Variational Learning Finds Flatter Solutions at the Edge of Stability

**Avrajit Ghosh**[1,*]    **Bai Cong**[2,3]    **Rio Yokota**[2]    **Saiprasad Ravishankar**[1]

**Rongrong Wang**[1]    **Molei Tao**[4]    **Mohammad Emtiyaz Khan**[3]    **Thomas Möllenhoff**[3]

[1]Michigan State University    [2]Institute of Science Tokyo
[3]RIKEN Center for AI Project    [4]Georgia Institute of Technology
[*]Work performed in part during internship at the RIKEN Center for AI Project.

## Abstract

Variational Learning (VL) has recently gained popularity for training deep neural networks. Part of its empirical success can be explained by theories such as PAC-Bayes bounds, minimum description length and marginal likelihood, but little has been done to unravel the implicit regularization in play. Here, we analyze the implicit regularization of VL through the Edge of Stability (EoS) framework. EoS has previously been used to show that gradient descent can find flat solutions and we extend this result to show that VL can find even flatter solutions. This result is obtained by controlling the shape of the variational posterior as well as the number of posterior samples used during training. The derivation follows in a similar fashion as in the standard EoS literature for deep learning, by first deriving a result for a quadratic problem and then extending it to deep neural networks. We empirically validate these findings on a wide variety of large networks, such as ResNet and ViT, to find that the theoretical results closely match the empirical ones. Ours is the first work to analyze the EoS dynamics of VL.

## 1 Introduction

Variational Learning (VL) has been used to perform deep learning from early on [Graves, 2011, Blundell et al., 2015] and recently also started to show good results at large scale. It has been shown to outperform state-of-the-art optimizers without any increase in the cost. For example, on ImageNet, VL substantially improves overfitting commonly seen in AdamW and for pretraining GPT-2 from scratch, VL achieves a lower validation perplexity than AdamW [Shen et al., 2024]. For low-rank fine-tuning of Llama-2 (7B), VL improves both accuracy (by $2.8\%$) and calibration (by $4.6\%$) [Cong et al., 2024, Li et al., 2025]. Moreover, VL methods explicitly derived by using PAC-Bayes bounds [Wang et al., 2023b, Zhang et al., 2024b] have shown consistent improvements over AdamW. All such results confirm the importance of VL for deep learning.

Despite these successes, the theoretical mechanisms behind the good performance of VL remain poorly understood. It is often assumed that simplistic Gaussian posteriors, such as those used currently for deep learning, may not be enough because they are poor approximations of the true posterior; lack of a good prior is another issue. Despite these concerns, VL shows good performance in practice. Theories such as minimum description length [Hinton and Van Camp, 1993, Hochreiter and Schmidhuber, 1997, Blier and Ollivier, 2018], PAC-Bayes [Dziugaite and Roy, 2017, Zhou et al., 2019, Lotfi et al., 2024, Alquier et al., 2024] and marginal likelihood [Smith and Le, 2017, Immer et al., 2021] can partially explain the success. In particular, PAC-Bayes theory provides a natural explanation for why flatter solutions may generalize better, see for instance the discussion in [Alquier et al., 2024, Section 3.3]. However these theories say little about the regularizing properties of the

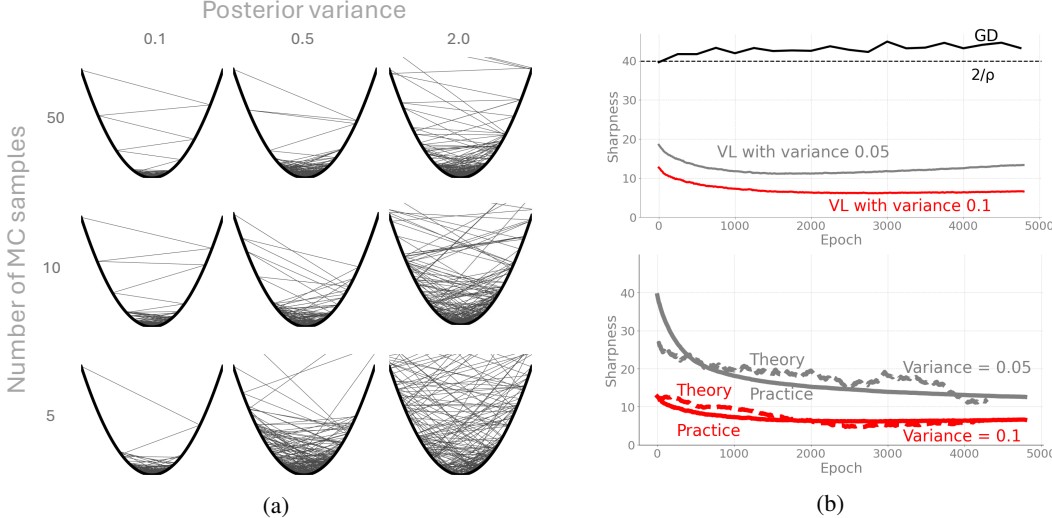

Figure 1: Panel (a): The left figure shows trajectory traces of VL on a quadratic problem with an isotropic variational posterior whose mean is learned but variance is set to a fixed value. The trajectory becomes more unstable as the posterior variance is increased and number of Monte-Carlo samples is decreased. We provide an exact expression to compute the *stability threshold* at which the iterations become unstable (Theorem 3.1). Panel (b): We show the validity of the threshold on neural network training. The right figure (top) shows this on CIFAR-10 for an MLP where VL achieves lower sharpness than GD when posterior variance is increased. The bottom figure shows that the sharpness (solid line) matches the stability threshold obtained by our theorem (dashed line).

learning algorithm. Similarly to deep learning, the presence of implicit regularization is likely also at play in VL, but few tools exist to unravel these effects.

In this work, we analyze the implicit regularization of VL algorithms at the Edge of Stability. The EoS analysis has previously been used to show that Gradient Descent (GD) with constant learning rate $\rho$ implicitly biases the trajectories towards flatter solutions, where the *sharpness* (defined as the operator norm of the loss Hessian) hovers around $2/\rho$. We extend this analysis to VL and show that sharpness can be further lowered by controlling the posterior covariance and the number of Monte-Carlo samples used to compute posterior expectations; see an illustration in Figure 1a. Similarly to the standard EoS technique, we first derive an exact expression for the stability threshold for VL on a quadratic problem and then propose extensions to general loss functions. We then empirically validate these finding on a wide variety of deep networks, including Multi-Layer Perceptron, ResNet, and Vision Transformers; see an example in Figure 1b. We observe similar results when posterior shape is automatically learned, for instance, by using diagonal covariance Gaussians and heavy-tailed posteriors. Code to replicate these results is available at https://github.com/Avra98/variationallearning-eos.

## 2 Theoretical Tools to Analyze Variational Learning

Variational Learning optimizes the variational reformulation of Bayesian learning [Zellner, 1988], where the goal is to find good approximations of the Gibbs distribution $\exp(-\ell(\boldsymbol{\theta}))/\mathcal{Z}$ with partition function $\mathcal{Z}$ over parameters $\boldsymbol{\theta} \in \mathbb{R}^d$. Specifically, we seek the closest distribution $q(\boldsymbol{\theta})$ in a set of distributions $\mathcal{Q}$ that minimizes the expected loss regularized by the entropy:

$$\underset{q \in \mathcal{Q}}{\arg\min} \ \mathbb{E}_{\boldsymbol{\theta} \sim q}[\ell(\boldsymbol{\theta})] - \mathcal{H}(q). \tag{1}$$

This is an instance of the maximum-entropy principle. The objective is equivalent to minimizing the KL divergence to the Gibbs posterior and naturally encourages higher-entropy (flatter) solutions due to the entropy $\mathcal{H}(q)$; for example, see the illustrative example in Khan and Rue [2023, Figure 1]. Similar arguments can also be made with the minimum-description-length principles [Hinton and Van Camp, 1993, Graves, 2011, Blier and Ollivier, 2018]. In practice, Variational Learning has started to show good results achieving generalization performance better than the state of the art

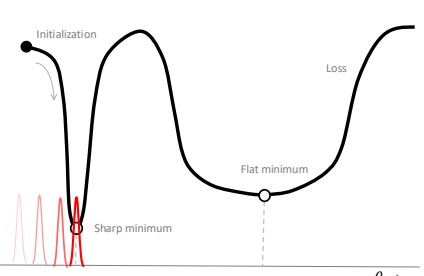
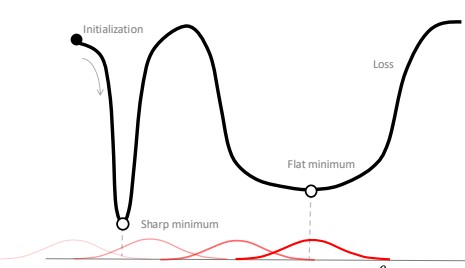

Figure 2: VL's mechanism for flatter minima: The posterior variance determines the minima's location. A small variance settles the posterior in a sharp minima (left), while a larger variance allows it to explore and find a flat minima (right).

optimizers across several tasks [Shen et al., 2024, Cong et al., 2024]. This matches with the intuition: Because the variational objective prefers wider distributions, we expect the posterior to be located in the region where the loss $\ell(\boldsymbol{\theta})$ is flatter, see Figure 2.

Despite this intuition, there is little work to analyze the implicit regularization that could help us understand how VL favors flatter regions and how we can control it. Existing theories do not sufficiently address this; a review of the related work is in Appendix A. We suspect the implicit regularization to be related to the shape of the posterior but currently there are no results explicitly characterizing this. VL is also closely connected to weight-noise or weight-perturbation methods where the goal is to optimize $\mathbb{E}_{q(\boldsymbol{\epsilon})}[\ell(\boldsymbol{\theta} + \boldsymbol{\epsilon})]$ with $q(\boldsymbol{\epsilon})$ being a fixed distribution to inject weight noise. This can be seen as a special case of variational learning where the shape of the posterior is fixed and only the location is learned. Multiple works [Zhu et al., 2019, Nguyen et al., 2019, Zhang et al., 2019, Jin et al., 2017, Simsekli et al., 2019] have analyzed generalization behavior of such weight-noise variational methods but, even for this simple case, there are no studies connecting them to the EoS results for GD. In this paper, we will address these gaps and provide both theoretical and empirical results regarding the edge of stability phenomenon of such a variational GD (VGD).

## 2.1 Edge of Stability for Gradient Descent

We will briefly review the EoS result for GD. The standard EoS literature relies on a result for a quadratic problem and then extends it to deep neural networks. For instance, consider the following

$$\text{quadratic loss:} \quad \ell(\boldsymbol{\theta}) = \frac{1}{2}\boldsymbol{\theta}^\top \mathbf{Q}\boldsymbol{\theta}, \quad \text{where} \quad \mathbf{Q} = \sum_{i=1}^{d} \lambda_i \mathbf{v}_i \mathbf{v}_i^T. \tag{2}$$

Here, $\mathbf{Q}$ is a positive definite matrix with $\lambda_i$ being its $i^{th}$ largest eigenvalue and $\mathbf{v}_i$ being the corresponding eigenvector. The following result states the condition under which one step of GD leads to a decrease in the loss.

**Lemma 2.1. (Descent Lemma)** *For a GD update $\boldsymbol{\theta}_{t+1} = \boldsymbol{\theta}_t - \rho\nabla\ell(\boldsymbol{\theta}_t)$ on the quadratic loss (2), the loss decreases at each step, that is, we have*

$$\ell(\boldsymbol{\theta}_{t+1}) - \ell(\boldsymbol{\theta}_t) \leq 0, \quad \textit{if and only if} \quad \lambda_i \leq \frac{2}{\rho} \quad \textit{for all } i. \tag{3}$$

This lemma implies that GD converges on a quadratic loss if all eigenvalues satisfy $\lambda_i < 2/\rho$. This is a different way of writing the standard condition that maximum eigenvalue $\lambda_1 < 2/\rho$. The result extends to any smooth $\ell(\boldsymbol{\theta})$ and for such cases, the condition implies that the learning rate $\rho < 2/\beta$, where $\beta = \sup_{\boldsymbol{\theta}} \|\nabla^2\ell(\boldsymbol{\theta})\|_2$ is the bound on the Hessian norm [Nesterov et al., 2018, Chapter 2]. This condition is *necessary and sufficient* for descent of GD on quadratic.

While the descent lemma is predictive of convergence of GD for smooth functions, deep learning exhibits a more complex behavior. When training deep neural networks with a constant learning rate $\rho$, the Hessian's operator norm (or sharpness) tends to settle around the value $2/\rho$. This phenomenon is

referred to as the *Edge of Stability*. Deep neural networks often operate at this edge and converge in a non-monotonic, unstable manner. A comprehensive empirical investigation of this phenomenon is given by Cohen et al. [2021], where two phases of training neural networks were noticed. In the first phase referred to as 'progressive sharpening', the Hessian's operator norm, $\|\nabla^2 \ell(\boldsymbol{\theta}_t)\|_2$ increases and slowly approaches $2/\rho$. This is followed by the second, EoS phase, where sharpness hovers around $2/\rho$ and the loss continues to decrease in an oscillatory fashion.

This phenomenon does not happen on a quadratic loss, but only for certain losses with nonzero third-order derivative. As shown by Damian et al. [2023], once sharpness reaches $2/\rho$, a local quadratic approximation is insufficient to capture the dynamics because the third order Taylor expansion term of the loss becomes significant. This cubic term represents the gradient of the sharpness, which serves as a negative feedback to counteract progressive sharpening and stabilize the sharpness around $2/\rho$ in GD. Instead of divergence, the iterates exhibit oscillatory or non-monotonic behavior even when the sharpness reaches $2/\rho$. This is the reason why $2/\rho$ is also called the 'Stability Threshold'. Increasing $\rho$ reduces the edge, which could then drive the iterates toward flatter minima that may generalize better. EoS analysis can help us understand such implicit regularization during training, especially for nonconvex problems.

Different optimizers have different stability thresholds which depends on the sharpness value $\|\nabla^2 \ell(\boldsymbol{\theta}_t)\|_2$. For instance, Sharpness Aware Minimization (SAM) leads to a different, smaller stability threshold [Long and Bartlett, 2024] compared to $2/\rho$ for GD. In this work, we show a similar result where VL has a smaller threshold than GD. We show this by deriving the stability threshold for a simpler quadratic problem first and analyze several factors such as posterior covariance and the number of posterior samples that influence the threshold. Then, we empirically demonstrate that similar results hold for the case when VL is used to train deep neural networks.

## 3   Stability Threshold for a Simple Case of Variational Learning

We start with a simple VL setting where the goal is to estimate a Gaussian $q(\boldsymbol{\theta}) = \mathcal{N}(\boldsymbol{\theta} \,|\, \boldsymbol{m}, \boldsymbol{\Sigma})$ whose mean $\boldsymbol{m}$ is unknown and covariance $\boldsymbol{\Sigma}$ is fixed. We assume $\boldsymbol{\Sigma} = \sigma^2 \mathbf{I}$ is an isotropic covariance matrix, with scalar variance $\sigma^2$. Khan and Rue [2023] [Section 1.3.1] show that this can be estimated by the following Variational GD algorithm:

$$\mathbf{m}_{t+1} \leftarrow \mathbf{m}_t - \rho \, \mathbb{E}_{\boldsymbol{\epsilon} \sim \mathcal{N}(\mathbf{0}, \boldsymbol{\Sigma})}[\nabla \ell(\mathbf{m}_t + \boldsymbol{\epsilon})]. \tag{4}$$

This algorithm can be implemented by the following version where the expectation is estimated by drawing $N_s$ Monte Carlo samples to approximate the expectation as follows

$$\mathbf{m}_{t+1}^{\boldsymbol{\epsilon}} \leftarrow \mathbf{m}_t - \rho \frac{1}{N_s} \sum_{i=1}^{N_s} \nabla \ell(\mathbf{m}_t + \boldsymbol{\epsilon_i}), \quad \boldsymbol{\epsilon}_i \sim \mathcal{N}(\mathbf{0}, \sigma^2 \mathbf{I}). \tag{5}$$

The updated iterate $\mathbf{m}_{t+1}^{\boldsymbol{\epsilon}}$ depends on the $N_s$ Monte Carlo samples $\boldsymbol{\epsilon} = [\boldsymbol{\epsilon}_1, \boldsymbol{\epsilon}_2, .., \boldsymbol{\epsilon}_{N_s}]$. Compared to the standard gradient descent, the update step in Variational GD (VGD), is determined by gradients averaged over a local neighborhood of perturbed weights. As a result, the update introduces two interacting effects influencing its stability threshold: (1) a *perturbation effect*, originating from the perturbation covariance $\boldsymbol{\Sigma}$, and (2) a *smoothing effect*, resulting from averaging gradients across $N_s$ Monte Carlo samples. Similarly to Lemma 2.1, we can derive the stability threshold for the above VGD and show that it is smaller than that of GD.

**Theorem 3.1.** *Consider the VGD update* (5) *on the quadratic loss* (2)*, then*

$$\mathbb{E}_{\boldsymbol{\epsilon}}[\ell(\mathbf{m}_{t+1}^{\boldsymbol{\epsilon}})] - \ell(\mathbf{m}_t) < 0 \quad if \quad \lambda_i < \frac{2}{\rho} \cdot \mathrm{VF}\left(\frac{N_s}{\sigma^2} \cdot c_{i,t}\right) \quad for\ all\ i, \tag{6}$$

*where* $c_{i,t} = (\lambda_i \mathbf{m}_t^\top \mathbf{v}_i)^2$, *and* $\mathrm{VF}(\cdot)$ *denotes the Variational Factor given by*

$$\mathrm{VF}(z) := \rho \cdot \sqrt{\frac{z}{3}} \cdot \sinh\left(\frac{1}{3} \mathrm{arcsinh}\left(\frac{3}{\rho}\sqrt{\frac{3}{z}}\right)\right). \tag{7}$$

See Appendix B for a proof where we also discuss the case where $\mathbf{Q}$ is low rank. The theorem is analogous to the descent lemma for GD and it states that the Variational GD (5) also decreases the

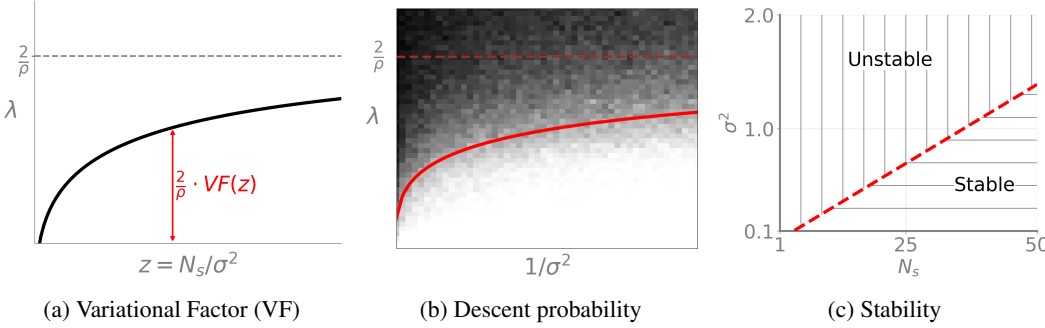

| (a) Variational Factor (VF) | (b) Descent probability | (c) Stability |

Figure 3: (a) Solid black curve shows the theoretical stability-threshold of VGD as a function of $N_s/\sigma^2$. The curve is clearly lower than the stability threshold of GD, shown with the horizontal dashed, gray line. (b) Empirical verification on a scalar quadratic problem with curvature $\lambda$ where we plot the empirically computed probability of descent for VGD runs with different values of $\sigma^2$. We show a heatmap for $(\lambda, 1/\sigma^2)$ values where lighter colors indicate higher probability of descent. We overlay the heatmap with the stability thresholds of VGD (red solid curve), clearly showing that theoretical limit shown in (6) matches the empirical probability. (c) The figure further includes $N_s$ and marks the region where a pair $(N_s, \sigma^2)$ will either lead to descent or not (marked with 'stable' and 'unstable' respectively).

expected loss over $\epsilon$ if each eigenvalue $\lambda_i$ is less than $2/\rho$ times a function called the Variational Factor (VF). The VF function is strictly less than 1, therefore the stability threshold is strictly less than $2/\rho$. Unlike GD, the condition here is only sufficient but not necessary. A necessary condition can also be derived but the one above is sufficient for this paper.

Figure 3a shows the stability threshold as a function of $z$ where it is clear that it always lower bounds $2/\rho$ (dashed horizontal line). The exact value of VF depends on its argument $z$ which in Theorem 3.1 mainly depends on the number of Monte-Carlo samples $N_s$ and posterior variance $\sigma^2$, but also on the loss-dependent constant $c_{it}$ which is essentially a function of $\lambda_i$ and $\mathbf{m}_t^\top \mathbf{v}_i$.

Theorem 1 guarantees a decrease in the expected loss. Below, we state another result to show that the actual loss also decreases with high probability if the expected loss decreases by a margin $\delta > 0$.

**Lemma 3.2.** *In the same setting as Theorem 3.1, when the expected loss at next iteration is smaller than the previous loss by some margin $\delta > 0$, that is,*

$$\mathbb{E}_{\boldsymbol{\epsilon}}[\ell(\mathbf{m}_{t+1}^{\boldsymbol{\epsilon}})] < \ell(\mathbf{m}_t) - \delta,$$

*then $\ell(\mathbf{m}_{t+1}^{\boldsymbol{\epsilon}}) - \ell(\mathbf{m}_t) < 0$ occurs with probability at least $1 - 2\exp\left(-c_1 \min\left\{\delta^2 N_s^2/c_2, \delta N_s/c_2\right\}\right)$, for constants $c_1, c_2 > 0$ depending only on $\rho$, $\mathbf{Q}$, and $\boldsymbol{\Sigma}$.*

The result also shows that the probability with which this happens increases with the number of Monte Carlo samples $N_s$.

So far, we show that the stability threshold for Variational GD is smaller than that of GD, but can we also ensure it to be smaller in practice? The answer is yes, and it can be done by controlling $\sigma^2$ and $N_s$. We will now demonstrate this on a 1D quadratic loss $\ell(m) = \frac{1}{2}\lambda m^2$. For such cases, GD with step-size $\rho$ descends whenever the curvature $\lambda$ is bounded by $2/\rho$, but we can show that VGD descends only for lower curvature values. To show this, we run VGD for many $(\lambda, \sigma)$ pairs. For each run, we use 10 random realizations of $\epsilon$, then record how often the loss decreases, and finally compute the approximate descent probability $\mathbb{P}\left(\ell(m_{t+1}^{\epsilon}) < \ell(m_t)\right)$.

Figure 3b shows this probability as a heatmap where lighter color indicate a higher probability of descent; a white pixel indicates a probability of 1 and a black one indicates a probability of 0. We overlay this with the solid red curve showing the theoretical stability-threshold of Variational GD as dictated by (6), that is, $\lambda = (2/\rho)\,\text{VF}(N_s/\sigma^2)$. The theoretical curve closely matches the transition boundary where probabilities transition from 1 to 0. The figure clearly shows that by increasing $\sigma^2$ we can reduce the stability threshold in practice too. Figure 3c further illustrates the effect of changing $N_s$, showing the regions where a $(N_s, \sigma^2)$ pair lead to descent (marked as 'stable') or otherwise (marked with 'unstable'). As expected, the relationship is linear and the same effect is obtained by either increasing $N_s$ or decreasing $\sigma^2$.

### 3.1 Nature of Stability in GD and VGD

The introduction of noise in VGD fundamentally changes the nature of its stability compared to the deterministic behavior of GD. The descent lemma for GD on a quadratic guarantees that the iterates are *asymptotically stable*, which is defined as follows:

**Definition 3.3** (Asymptotic Stability Lyapunov [1992], Chapter 2). *Let $\boldsymbol{\theta}^*$ be the minimum. An iterate $\boldsymbol{\theta}_k$ is asymptotic stable if it is Lyapunov stable and there also exists a $\delta > 0$ such that if the algorithm starts within a $\delta$-neighborhood of the fixed point, the iterate will converge to the fixed point. Formally, if $\|\boldsymbol{\theta}_k - \boldsymbol{\theta}^*\| < \delta$ for a finite $k$, then:*

$$\lim_{k \to \infty} \|\boldsymbol{\theta}_k - \boldsymbol{\theta}^*\| = 0. \tag{8}$$

GD on a quadratic is asymptotically stable because the condition learning-rate $\rho < 2/\lambda_1$ ensures that the iteration matrix $(\mathbf{I} - \rho\mathbf{Q})$ has all eigenvalues less than 1, guaranteeing convergence of the iterates to its fixed point. Analogously VGD, the iterates are *Stochastically* Stable which is defined as follows

**Definition 3.4** (Stochastic Stability Kushner [1972], Chapter 2). *Let $\boldsymbol{\theta}^*$ be the minimum. The VGD iterates $\boldsymbol{\theta}_t^{\boldsymbol{\epsilon}}$ is said to be stochastically stable in the mean-square sense if there exists a constant $C > 0$ such that for any initial point $\boldsymbol{\theta}_0$, the iterates $\boldsymbol{\theta}_t^{\boldsymbol{\epsilon}}$ satisfy:*

$$\mathbb{E}_{\boldsymbol{\epsilon}}\left[\|\boldsymbol{\theta}_t^{\boldsymbol{\epsilon}} - \boldsymbol{\theta}^*\|^2\right] \leq C\|\boldsymbol{\theta}_0 - \boldsymbol{\theta}^*\|^2, \quad \text{for all } t > 0. \tag{9}$$

This form of stability ensures that the iterates remain bounded in the mean-square error sense, preventing them from diverging. Practically, this corresponds to the behavior where the distribution of the iterates converges to a stationary distribution, rather than the iterates themselves converging to a single fixed point as in asymptotic stability. The condition for probabilistic descent, established in Theorem 3.1, is what characterizes this stable behavior. By ensuring the loss decreases on average, it prevents the divergence of the iterates, confining them to a stable random walk that converges in distribution to a stationary state around the minimum. Accordingly, any reference to "stability" or a "stability threshold" throughout for VGD in this paper refers to this notion of stochastic stability.

### 3.2 Comparison with Regularization Effect of SGD

Our work differs fundamentally from studies on the regularization effects of mini-batch noise in SGD, such as Wu et al. [2022], Zhu et al. [2019], Wu et al. [2018], Ibayashi and Imaizumi [2023], Mulayoff and Michaeli [2024], which analyze gradient noise and its role in promoting flatter solutions. In contrast, VGD introduces noise directly in the weights, inducing a structured and anisotropic form of gradient noise shaped by the curvature of the loss landscape, something not captured by existing SGD analyses. While most SGD-based studies assume Gaussian gradient noise, despite empirical evidence of heavy-tailed behavior [Gurbuzbalaban et al., 2021, Nguyen et al., 2019], we show that for the quadratic loss, Gaussian perturbations in weights lead to Gaussian-distributed gradients, that is, $\hat{\mathbf{g}} \sim \mathcal{N}(\nabla\ell(\mathbf{m}_t), \frac{1}{N_s}\mathbf{Q}\boldsymbol{\Sigma}\mathbf{Q})$, where the covariance is amplified along sharper directions. This formulation requires no assumptions on the shape of gradient noise (unlike, e.g., Lee and Jang [2023]). To complement our work, we further perform an empirical study using weight perturbations drawn from a heavy-tailed distribution in deep neural networks.

## 4 Experiments on Deep Neural Networks

Similarly to the GD case, we expect the stability threshold for the quadratic to serve as an EoS limit. That is, we expect that, when using variational learning for deep neural networks, the sharpness should hover around the stability limit. By controlling the covariance shape and number of Monte Carlo samples we can stere optimization into regions where sharpness is much lower than GD. The following hypothesis formally states this intuition which we verify through extensive experiments.

**Hypothesis 1.** *For a twice-differentiable loss $\ell(\mathbf{m})$ optimized using the update rule (5), the top Hessian eigenvalue $\|\nabla^2\ell(\mathbf{m}_t)\|_2$ hovers around the stability threshold,*

$$\frac{2}{\rho} \cdot \text{VF}\left(\frac{N_s}{\sigma^2} \cdot c_{1,t}\right),$$

*where $c_{1,t} = (\mathbf{v}_{1,t}^\top \nabla\ell(\mathbf{m}_t))^2$ and $\mathbf{v}_{1,t}$ denotes the top eigenvector of the Hessian $\nabla^2\ell(\mathbf{m}_t)$, and $\nabla\ell(\mathbf{m}_t)$ is the gradient evaluated at $\mathbf{m}_t$.*

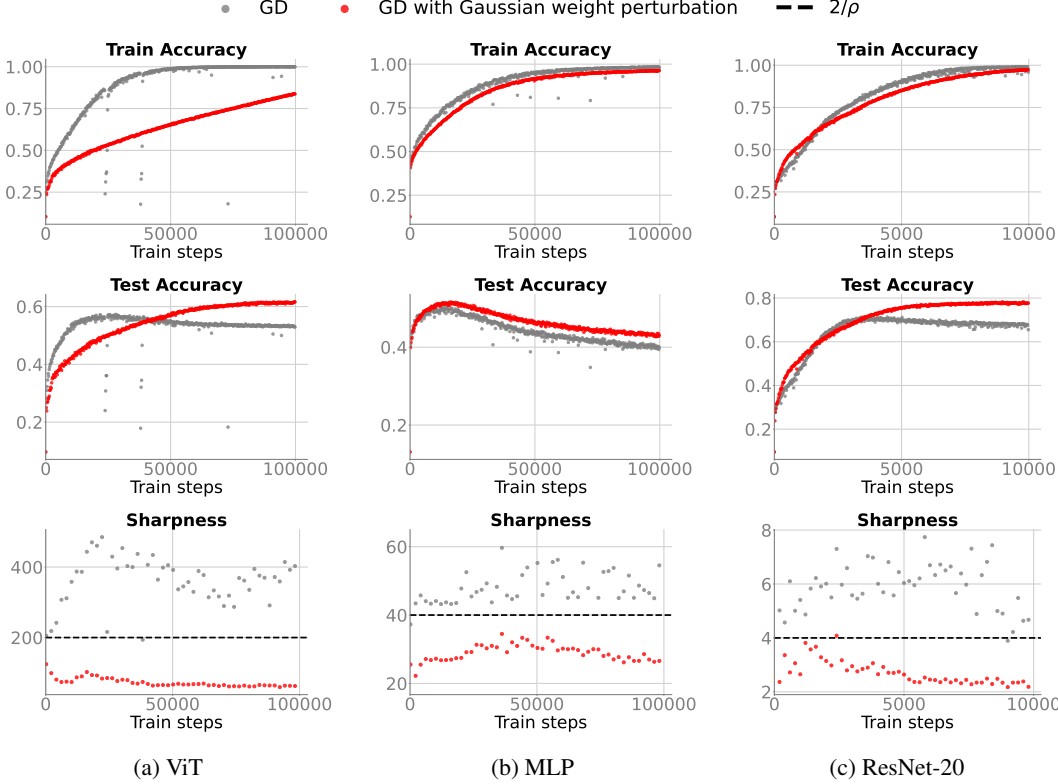

(a) ViT  (b) MLP  (c) ResNet-20

Figure 4: Smaller sharpness corresponds to higher test accuracy for network architectures trained on CIFAR-10. Panels show (a) ViT, (b) MLP, and (c) ResNet-20. Full batch GD with Variational GD achieves lower sharpness and better test accuracy.

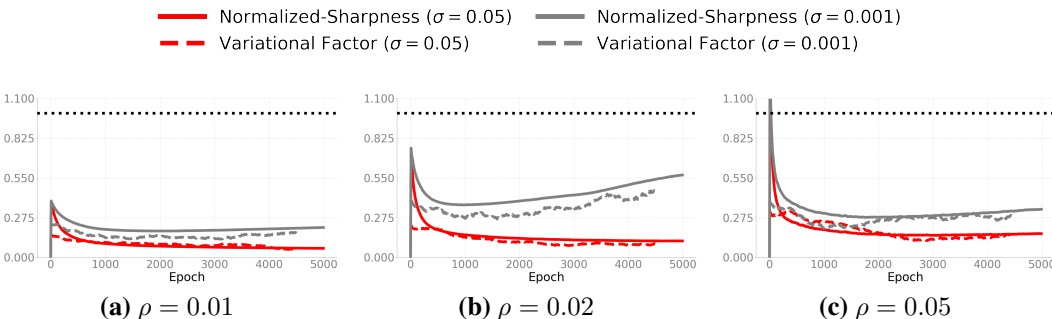

**(a)** $\rho = 0.01$  **(b)** $\rho = 0.02$  **(c)** $\rho = 0.05$

Figure 5: Normalized Sharpness $\|\nabla^2 \ell(\mathbf{m}_t)\|_2/(2/\rho)$ hovers around the Variational Factor in MLP.

To validate the hypothesis, we conduct extensive experiments on standard architectures (MLPs, ResNets) and modern ones such as Vision Transformers. In Figure 4, we plot the full training dynamics of Variational GD, including test accuracy, training accuracy, and sharpness. Across all three architectures on the CIFAR-10 classification task using MSE loss, Variational GD with isotropic Gaussian noise consistently achieves lower sharpness and higher test accuracy compared to GD.

In Figure 5, we plot the normalized sharpness ($\|\nabla^2 \ell(\mathbf{m}_t)\|_2$ divided by $2/\rho$) for MLPs, evaluated at each training step, alongside the corresponding Variational Factor (VF) for various learning rates $\rho$ and posterior variances $\sigma^2$. The results support Hypothesis 1, showing that the normalized sharpness closely tracks the predicted VF across settings. This further validates the use of a local quadratic approximation to analyze stability dynamics, consistent with several prior works. Unlike the stability analysis for GD on a quadratic where the threshold $2/\rho$ is constant, the stability condition in VGD

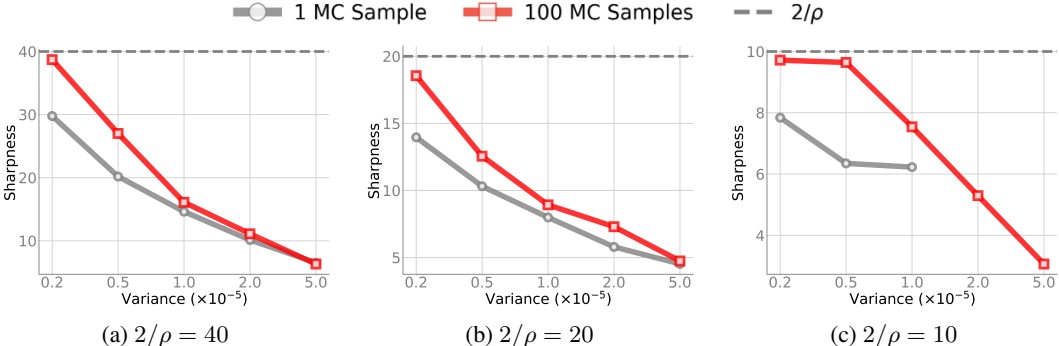

Figure 6: Sharpness of the final iterate for an MLP trained on CIFAR-10 for variational GD across learning rate $\rho$, noise covariance $\boldsymbol{\Sigma}$, and posterior samples $N_s$. Higher variance and smaller samples lead to lower sharpness. For panel (c), training did not converge for variance values 2 and 5, therefore not shown in the plot.

is dynamic since it depends on the gradient evaluated at the mean. This explains the fluctuation of the stability threshold across iterations in VGD. In the Appendix (Figure 14), we demonstrate that a similar phenomenon holds for ResNet-20. Next, we isolate and examine the roles of the posterior variance and the number of samples in determining this stability threshold.

**Perturbation effect of Gaussian variance:** In Figure 6, we present experiments on VGD for a classification task using a multi-layer perceptron (MLP), where we plot the sharpness of the final iterate as a function of the variance $\sigma^2$ of an isotropic Gaussian. For three different learning rates, $\rho = 0.05$, 0.1, and 0.2, we run VGD with different $\sigma^2$. Across all learning rates $\rho$ and numbers of posterior samples $N_s$, we observe that larger variance consistently leads to lower sharpness. This exactly aligns with Hypothesis 1, larger perturbations reduce the stability threshold, promoting escape from sharper regions of the loss landscape.

**Smoothing effect of posterior samples $N_s$:** Larger posterior sample size $N_s$ reduces the variance of the perturbed gradient estimator $\hat{\mathbf{g}} = \frac{1}{N_s} \sum_{i=1}^{N_s} \nabla\ell(\mathbf{m}_t + \boldsymbol{\epsilon}_i)$, which satisfies $\hat{\mathbf{g}} \sim \mathcal{N}(\nabla\ell(\mathbf{m}_t), \frac{1}{N_s}\mathbf{Q}\boldsymbol{\Sigma}\mathbf{Q})$ under a quadratic loss. As $N_s$ decreases, the variance increases (scaling as $1/N_s$), lowering the stability threshold and enabling escape from sharper regions. This effect is shown in Figure 6, where smaller $N_s$ consistently yields lower sharpness across learning rates and variances. Additional discussion on the role of posterior samples, is provided in Appendix C.

## 4.1 Edge of Stability under Heavy-Tailed Noise

In variational gradient descent (VGD), the stability threshold depends not only on the perturbation covariance and sample size, but also on the posterior's tail behavior. We empirically investigate this by drawing weight perturbations from a Student-t distribution with degrees of freedom $\alpha$, where smaller $\alpha$ induces heavier tails and larger deviations in the perturbed gradients. The distribution becomes heavier-tailed as $\alpha$ decreases, with the Gaussian recovered as $\alpha \to \infty$. In Figure 7, we train an MLP on CIFAR-10 and observe that heavier-tailed perturbations yield lower sharpness and better test accuracy. Similar trends are confirmed for ResNet-20 and ViT in Appendix F. While a theoretical analysis under heavy-tailed noise is challenging, our results highlight that the posterior shape plays a critical role in generalization.

## 4.2 Adaptive Edge of Stability and Variational Online Newton Methods

Recent work by Cohen et al. [2022] studies the dynamics of adaptive gradient methods, introducing the concept of Adaptive Edge of Stability (AEoS). In this section, we will show that a similar phenomenon happens in natural-gradient variational learning.

Cohen et al. [2022] show that for adaptive gradient methods instead of sharpness $\|\nabla^2\ell(\mathbf{m}_t)\|_2$, a modified quantity, termed preconditioned sharpness $\|\text{diag}(\mathbf{p}_t)^{-1}\nabla^2\ell(\mathbf{m}_t)\|_2$ hovers around $2/\rho$, where $\mathbf{p}_t$ is a preconditoner. Adaptive gradient methods differ from standard gradient descent as the

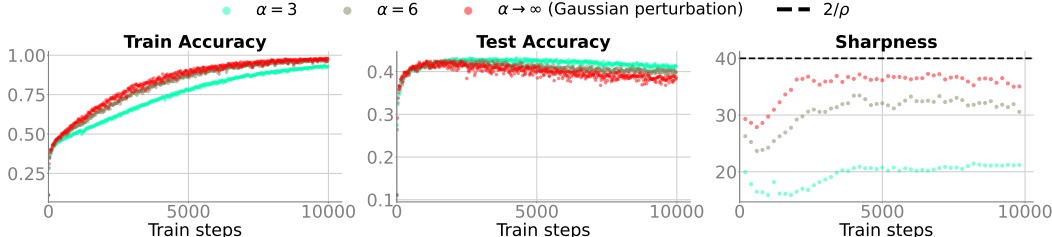

Figure 7: Sharpness dynamics with noise injection from a heavy tailed Student t posterior parameterized by $\alpha$. Perturbations from heavier tails (smaller $\alpha$) lead to smaller sharpness.

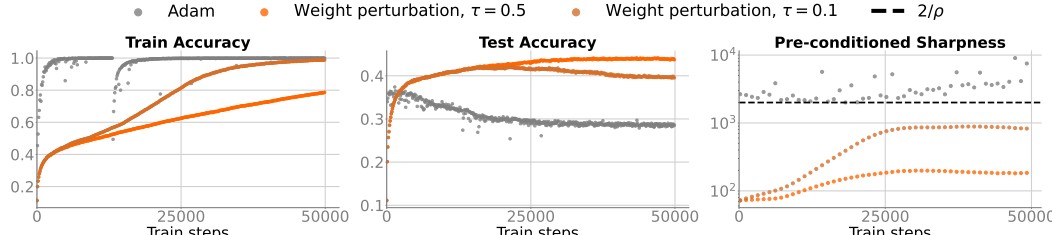

Figure 8: Preconditioned sharpness for Adam and IVON across temperatures $\tau$. Smaller $\tau$ shrinks the posterior and yields larger preconditioned sharpness.

former can adapt their preconditioner $\mathbf{P}_t$ and move into high curvature regions. For example in Adam, a preconditioner $\mathbf{p}_t$ is updated in an exponential moving average (EMA) fashion with parameter $\beta_2$:

$$\mathbf{v}_{t+1} = \beta_2 \mathbf{v}_t + (1 - \beta_2)\nabla\ell(\mathbf{m}_t)^2, \ \ \mathbf{p}_{t+1} = \sqrt{\frac{\mathbf{v}_{t+1}}{1 - \beta_2^{t+1}}}, \ \ \mathbf{m}_{t+1} = \mathbf{m}_t - \rho\,\nabla\ell(\mathbf{m}_t)/\mathbf{p}_{t+1}. \quad (10)$$

Here, all operations such as squaring or division of vectors are performed elementwise.

Natural-gradient VL methods which learn a complete Gaussian posterior $q_t = \mathcal{N}(\mathbf{m}_t, \mathbf{P}_t^{-1})$ take a similar form to the above adaptive gradient methods. An instance of this is the Variational Online Newton (VON) update rule [Khan and Rue, 2023, Eq. 12],

$$\mathbf{m}_{t+1} \leftarrow \mathbf{m}_t - \rho\,\mathbf{P}_{t+1}^{-1}\,\mathbb{E}_{q_t}[\nabla_\theta\ell(\boldsymbol{\theta})] \quad \text{and} \quad \mathbf{P}_{t+1} \leftarrow (1 - \beta_2)\,\mathbf{P}_t + \beta_2\,\mathbb{E}_{q_t}[\nabla_\theta^2\ell(\boldsymbol{\theta})]. \quad (11)$$

Here, the posterior covariance $\mathbf{P}_t^{-1}$ is learned using the loss Hessians. The variational GD in Eq. (5) corresponds to the special case where $\mathbf{P}_t$ is fixed across iterations. Adaptive optimizers such as Adam, RMSProp, and Adadelta can be seen as special cases of VON, as shown in [Khan and Rue, 2023, Section 4.2].

Here, we study the AEoS for a recent large-scale implementation of Equation (11) by Shen et al. [2024] called IVON (Improved Variational Online Newton). There, the update for the preconditioner is approximated using Stein's identity to estimate the Hessian from $N_s$ samples:

$$\mathbb{E}_{q_t}[\mathrm{diag}(\nabla_\theta^2\ell(\theta))] \approx \frac{1}{N_s}\sum_{i=1}^{N_s}\left(\nabla\ell(\boldsymbol{\theta}_i)\cdot\frac{\boldsymbol{\theta}_i - \mathbf{m}_t}{\sigma^2}\right), \quad \boldsymbol{\theta}_i \sim q_t. \quad (12)$$

In Figure 8, we compare the preconditioned sharpness of IVON and Adam on MLPs by varying the temperature $\tau$, which linearly scales covariance of the posterior distribution $q_t$, see [Shen et al., 2024, Algorithm 1, line 8]. We observe that IVON consistently yields lower preconditioned sharpness than the $2/\rho$ threshold typically observed in Adam. We conjecture the lower sharpness is due to the noise injection and smoothing effects in estimating both the gradient and curvature. Additional comparisons for ResNet-20 and ViT are provided in Appendix F. Computing the exact stability threshold for IVON is nontrivial, as it requires a joint analysis of the coupled dynamics in Eq. 11, and an interesting direction for future work.

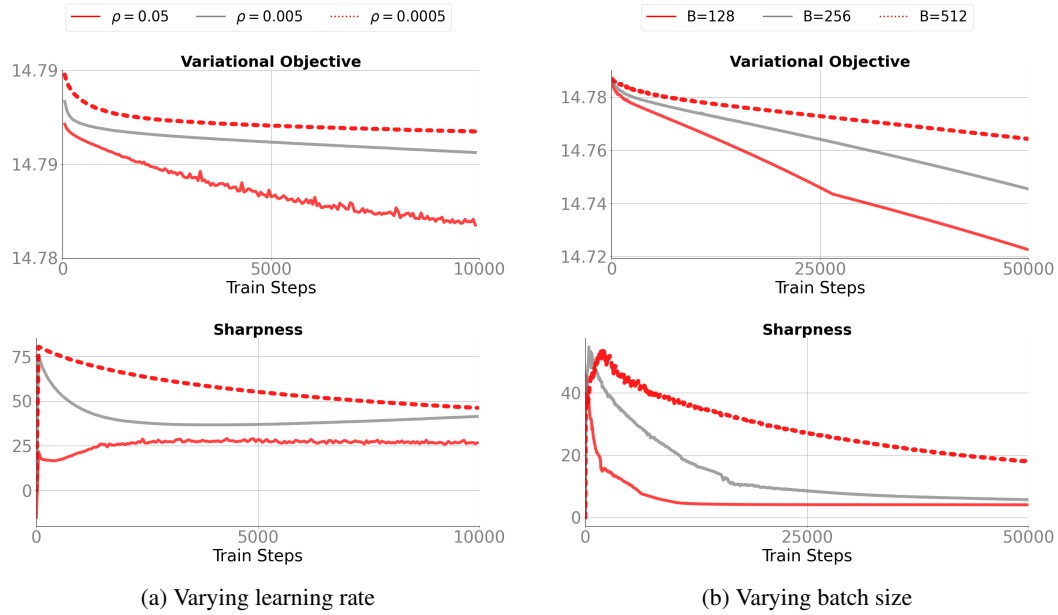

|     |     |
| --- | --- |
| (a) Varying learning rate | (b) Varying batch size |

Figure 9: Minimizing the variational objective depends on the choice of hyperparameter and its implicit regularization effect, visible in both the objective value and sharpness reduction.

### 4.3 Effect of Batch Size

For nonconvex problems such as deep learning, minimizing the variational objective (1) is not, by itself, sufficient to guarantee flat minima or a low objective value. The choice of optimization dynamics and hyperparameters play a crucial role as well. We demonstrate this by running IVON with mini-batching across varying learning rates $\rho$ and batch sizes $B$. As shown in Figure 9, larger learning rates and smaller batch sizes consistently lead to better local minima of the variational objective. The present work offers an explanation why large learning rates work better, but we also speculate that smaller batch sizes encourages broader posteriors in flatter minima allowing a smaller objective value. These results further highlight the importance of choice of hyperparameter and optimizers in variational learning. Poor performance of variational methods as claimed in the literature may not be due to flaws in the variational formulation itself but due to unfavorable optimization dynamics or choices of hyperparameters.

## 5    Conclusion and Discussion

In this work, we study the regularization effect in Variational Learning (VL) that enables it to find solutions with better generalization than Gradient Descent (GD). We show that the sharpness dynamics in VL can be accurately tracked through its instability mechanism under a local quadratic approximation of the loss. We argue that to fully explain generalization in VL, one must look beyond theoretical frameworks like PAC-Bayes bounds and instead understand the optimization dynamics and the implicit regularization they induce. We hope our work takes a positive step in this direction and motivates further investigation into the role of training dynamics in Bayesian deep learning.

### Acknowledgements

This work is supported by the Bayes duality project, JST CREST Grant Number JPMJCR2112. A.Ghosh and R.Wang acknowledge support from NSF Grant CCF-2212065. S.Ravishankar acknowledge support from NSF CAREER CCF-2442240 and NSF Grant CCF-2212065. M.Tao is grateful for partial supports by NSF Grant DMS-1847802, Cullen-Peck Scholarship, and Emory-GT AI.Humanity Award. We acknowledge Zhedong Liu for his help with the experiments on heavy-tailed distribution and Pierre Alquier for helpful discussions.

## Impact Statement

This work improves understanding of how learning algorithms can find solutions that generalize better. It may lead to more reliable and robust AI systems. While primarily theoretical, the insights could support better model training in practical settings. We see minimal risk of misuse, but encourage responsible use in sensitive applications.

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

# Supplementary Material

The supplementary material contains the following appendix section.

## A   Related Works

**Variational Learning**

Given a prior and a likelihood, variational inference approximates the posterior by optimizing the evidence lower bound within a family of posterior distribution. The ELBO is optimized by natural gradient descent [Amari, 1998], in the hope that steepest descent induced by the KL divergence is a better metric to compare probability distributions, this gives to rise of a class of algorithms called Natural Gradient Variational Learning. Several works have attempted to apply this to deep learning [Khan et al., 2018, Osawa et al., 2019, Lin et al., 2020]. Recently, Shen et al. [2024] provide an improved version of varlational Online Newton that largely scales and obtains state of the art accuracy and uncertainty at identical cost as Adam. A step of VL includes weight perturbation based on noise injection from the posterior distribution which has similarities to noise injection.

**Regularization Effect of Noise**

Injecting noise into the parameters within gradient descent has several desirable features such as escaping saddle points [Jin et al., 2017, Reddi et al., 2018] and local minima Zhu et al. [2019], Nguyen et al. [2019]. In fact it has been widely observed empirically that the SGD noise has an important role to play to find flat minima. Noise with larger scale [Zhu et al., 2019, Nguyen et al., 2019, Zhang et al., 2019, Smith et al., 2020, Wei and Schwab, 2019] and heavy-tail [Simsekli et al., 2019, Panigrahi et al., 2019, Nguyen et al., 2019, Wang et al., 2022a] drives the optimization trajectories towards flatter minima. Orvieto et al. [2023] demonstrated through stochastic Taylor expansion that injecting Gaussian noise into parameters before a gradient step implicitly regularizes by penalizing the curvature of the loss, while Orvieto et al. [2022] showed that anticorrelated noise in Perturbed Gradient Descent (PGD) specifically penalizes the trace of the Hessian. Zhang et al. [2024a] showed that injecting noise in opposite directions to the weight space leads to a better regularization of the trace of the Hessian.

**Gradient Descent at Edge of Stability**

Cohen et al. [2021], building on the work of Jastrzebski et al. [2020] empirically showed that for full batch Gradient Descent (GD) with constant step-size ($\rho$), the operator norm of the Hessian (also termed as sharpness) settles in a neighborhood of $2/\rho$. This threshold is termed as *edge of stability* (EoS) because gradient descent on a quadratic only converges if the sharpness is below $2/\rho$. Strikingly in complex neural network landscapes, sharpness settling around the stability limit $2/\rho$ (instead of diverging) indicates that presence of a self-stabilization mechanism Damian et al. [2023] (which is absent in a quadratic) that regularizes the sharpness near $2/\rho$. The common occurrence of this phenomenon across various tasks, architectures and initialization has inspired substantial research on edge of stability. For example EoS has been studied across several non-convex optimization problems [Wang et al., 2022b, 2023a, Zhu et al., 2023, Chen and Bruna, 2023, Agarwala et al., 2023, Arora et al., 2022, Lyu et al., 2022, Ahn et al., 2024, Wu et al., 2024, Ghosh et al., 2025, Even et al., 2024, Chen et al., 2024, Kreisler et al., 2023, Kalra et al., 2025, Zhu et al., 2024] and across other descent based optimizers such as SGD [Lee and Jang, 2023, Andreyev and Beneventano, 2024], momentum [Phunyaphibarn et al., 2024], sharpness aware minimization (SAM) Agarwala and Dauphin [2023], Long and Bartlett [2024] and Adam/RMSProp [Cohen et al., 2022, 2025].

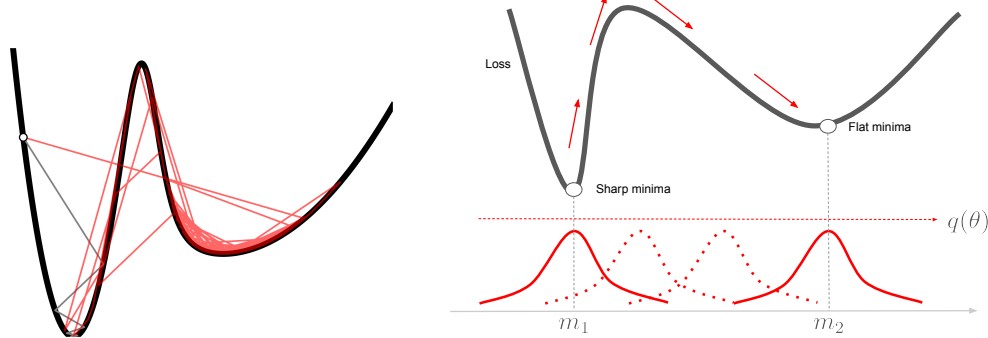

Figure 10: Shows the mechanism through which VL finds flatter minima. GD (gray iterates) get stuck in a local sharp minima whereas VL (red iterates) escapes to a flatter minima.

**Stability of GD with Noise**

Wu et al. [2018] analyze the *linear dynamical stability* of SGD, demonstrating that the batch size and the gradient covariance matrix impose an additional constraint, requiring the Hessian operator norm to be smaller than $2/\rho$. Wu et al. [2022] extend this work by assuming alignment between the gradient covariance matrix and the Fisher matrix, arguing that noise concentrates in the sharp directions of the landscape. Although neither study analyzes the EoS limit for SGD, they were instrumental in understanding the local stability of SGD near a global minimum. Lee and Jang [2023], Andreyev and Beneventano [2024] investigate the dynamics of SGD at EoS and instead propose that a different metric, called *mini-batch aware sharpness*, must be smaller than $2/\rho$ for stability. Since the actual sharpness is less than this mini-batch aware sharpness, sharpness itself must be smaller than $2/\rho$. However, in our work, the perturbation is in the weight-space and not on the gradient.

## B    Proof of Theorem 1

First, we prove a sufficient condition for descent for variational GD in Theorem B. The following theorem states the condition on the eigenspectrum for which descent takes place in expectation.

**Theorem 3.1.** *Consider the VGD update* (5) *on the quadratic loss* (2), *then*

$$\mathbb{E}_{\boldsymbol{\epsilon}}[\ell(\mathbf{m}_{t+1}^{\boldsymbol{\epsilon}})] - \ell(\mathbf{m}_t) < 0 \quad if \quad \lambda_i < \frac{2}{\rho} \cdot \text{VF}\left(\frac{N_s}{\sigma^2} \cdot c_{i,t}\right) \quad for\ all\ i, \tag{6}$$

*where* $c_{i,t} = (\lambda_i \mathbf{m}_t^\top \mathbf{v}_i)^2$, *and* $\text{VF}(\cdot)$ *denotes the Variational Factor given by*

$$\text{VF}(z) := \rho \cdot \sqrt{\frac{z}{3}} \cdot \sinh\left(\frac{1}{3}\operatorname{arcsinh}\left(\frac{3}{\rho}\sqrt{\frac{3}{z}}\right)\right). \tag{7}$$

*Proof.* We analyze the stability of the weight-perturbed gradient descent (GD) update applied to the quadratic loss

$$\ell(\boldsymbol{m}) = \frac{1}{2}\boldsymbol{m}^\top \mathbf{Q}\boldsymbol{m}, \tag{13}$$

where $\mathbf{Q} \in \mathbb{R}^{d \times d}$ is positive definite. The update rule is given by

$$\boldsymbol{m}_{t+1} \leftarrow \boldsymbol{m}_t - \rho\,\hat{\boldsymbol{g}}, \quad \text{where} \quad \hat{\boldsymbol{g}} := \frac{1}{N_s}\sum_{i=1}^{N_s}\nabla\ell(\boldsymbol{m}_t + \boldsymbol{\epsilon}_i), \quad \boldsymbol{\epsilon}_i \sim \mathcal{N}(\mathbf{0}, \boldsymbol{\Sigma}). \tag{14}$$

Since $\ell$ is quadratic, its gradient at perturbed point satisfies

$$\nabla\ell(\boldsymbol{m}_t + \boldsymbol{\epsilon}) = \nabla\ell(\boldsymbol{m}_t) + \mathbf{Q}\boldsymbol{\epsilon}. \tag{15}$$

Therefore, if $\epsilon \sim \mathcal{N}(\mathbf{0}, \mathbf{\Sigma})$, then
$$\nabla\ell(\boldsymbol{m}_t + \epsilon) \sim \mathcal{N}\left(\nabla\ell(\boldsymbol{m}_t), \mathbf{Q}\mathbf{\Sigma}\mathbf{Q}\right), \tag{16}$$
by linear transformation of Gaussian variables.

As $\hat{\boldsymbol{g}}$ is the average of $N_s$ i.i.d. samples from this distribution, it follows that
$$\hat{\boldsymbol{g}} \sim \mathcal{N}\left(\nabla\ell(\boldsymbol{m}_t), \frac{1}{N_s}\mathbf{Q}\mathbf{\Sigma}\mathbf{Q}\right). \tag{17}$$

Now, the Taylor expansion (exact) of the quadratic loss around $\boldsymbol{m}_t$ gives
$$\ell(\boldsymbol{m}_{t+1}) = \ell(\boldsymbol{m}_t) + \nabla\ell(\boldsymbol{m}_t)^\top(\boldsymbol{m}_{t+1} - \boldsymbol{m}_t) + \frac{1}{2}(\boldsymbol{m}_{t+1} - \boldsymbol{m}_t)^\top\mathbf{Q}(\boldsymbol{m}_{t+1} - \boldsymbol{m}_t). \tag{18}$$

Substituting the update rule (14) into (18), we get
$$\ell(\boldsymbol{m}_{t+1}) - \ell(\boldsymbol{m}_t) = -\rho\nabla\ell(\boldsymbol{m}_t)^\top\hat{\boldsymbol{g}} + \frac{\rho^2}{2}\hat{\boldsymbol{g}}^\top\mathbf{Q}\hat{\boldsymbol{g}}. \tag{19}$$

We now observe that the change in loss, denoted by $\Delta\ell := \ell(\boldsymbol{m}_{t+1}) - \ell(\boldsymbol{m}_t)$, is a random variable due to the stochasticity in the update. Using the expression from Equation (19), we can expand $\Delta\ell$ into a deterministic (mean) and a stochastic (fluctuation) component ($R$).

$$\Delta\ell = \ell(\mathbf{m}_{t+1}) - \ell(\mathbf{m}_t) = -\rho\,\mathbf{g}^\top\hat{\mathbf{g}} + \frac{\rho^2}{2}\hat{\mathbf{g}}^\top\mathbf{Q}\hat{\mathbf{g}} \tag{20}$$

$$= -\rho\,\mathbf{g}^\top(\mathbf{g} + \boldsymbol{\xi}) + \frac{\rho^2}{2}(\mathbf{g} + \boldsymbol{\xi})^\top\mathbf{Q}(\mathbf{g} + \boldsymbol{\xi}) \tag{21}$$

$$= \underbrace{-\rho\|\mathbf{g}\|^2 + \frac{\rho^2}{2}\mathbf{g}^\top\mathbf{Q}\mathbf{g} + \frac{\rho^2}{2}\mathbb{E}[\boldsymbol{\xi}^\top\mathbf{Q}\boldsymbol{\xi}]}_{\mathbb{E}[\Delta\ell]} + \underbrace{\left(-\rho\,\mathbf{g}^\top\boldsymbol{\xi} + \rho^2\,\mathbf{g}^\top\mathbf{Q}\boldsymbol{\xi} + \frac{\rho^2}{2}\left(\boldsymbol{\xi}^\top\mathbf{Q}\boldsymbol{\xi} - \mathbb{E}[\boldsymbol{\xi}^\top\mathbf{Q}\boldsymbol{\xi}]\right)\right)}_{R}$$
$$\tag{22}$$

We write the perturbed gradient as $\hat{\mathbf{g}} = \mathbf{g} + \boldsymbol{\xi}$, where $\boldsymbol{\xi} \sim \mathcal{N}\left(\mathbf{0}, \frac{1}{N_s}\mathbf{Q}\mathbf{\Sigma}\mathbf{Q}\right)$. Here $\mathbf{g}$ denotes the gradient evaluated at the posterior mean, that is, $\nabla\ell(\boldsymbol{m}_t)$. This expresses the update as the sum of a deterministic component $\mathbf{g}$ and a random fluctuation $\boldsymbol{\xi}$.

To ensure that the update leads to descent on average, we compute the expected change in loss and enforce the stability condition $\mathbb{E}[\Delta\ell] < 0$ with respect to the curvature.

Taking the expectation of the loss change derived in Equation (19), we obtain:
$$\mathbb{E}[\Delta\ell] = -\rho\mathbf{g}^\top(\mathbf{I} - \frac{\rho}{2}\mathbf{Q})\mathbf{g} + \frac{\rho^2}{2N_s}\operatorname{Tr}\left(\mathbf{\Sigma}\mathbf{Q}^3\right) < 0 \tag{23}$$

Since the Hessian $\mathbf{Q}$ is symmetric, it admits an eigendecomposition $\mathbf{Q} = \sum_{i=1}^d \lambda_i\mathbf{v}_i\mathbf{v}_i^\top$, where $(\lambda_i, \mathbf{v}_i)$ are the eigenvalue-eigenvector pairs. We expand the expression in the eigenbasis of $\mathbf{Q}$. Note the following identities:

- $\mathbf{g}^\top\mathbf{g} = \sum_{i=1}^d (\mathbf{v}_i^\top\mathbf{g})^2$,

- $\mathbf{g}^\top\mathbf{Q}\mathbf{g} = \sum_{i=1}^d \lambda_i(\mathbf{v}_i^\top\mathbf{g})^2$,

- $\operatorname{Tr}(\mathbf{\Sigma}\mathbf{Q}^3) \leq \sum_{i=1}^d \sigma_i\lambda_i^3$  (Von Neuman's trace inequality).

Thus, we obtain an upper bound on the expected descent:
$$\mathbb{E}[\ell(\mathbf{m}_{t+1})] - \ell(\mathbf{m}_t) \leq \sum_{i=1}^d \left[-\rho(\mathbf{v}_i^\top\mathbf{g})^2 + \frac{\rho^2}{2}\lambda_i(\mathbf{v}_i^\top\mathbf{g})^2 + \frac{\rho^2}{2N_s}\sigma_i\lambda_i^3\right]$$
$$=: \sum_{i=1}^d f(\lambda_i, \mathbf{v}_i).$$

The inequality is an equality for an isotropic Gaussian $\mathbf{\Sigma}$. For anisotropic covariance matrix, the tightness of this inequality depends on the alignment of the Hessian and the covariance matrix.

To ensure descent in expectation, we require $f(\lambda_i, \mathbf{v}_i) < 0$ for all $i$. We note that this is a sufficient condition for descent to take place. We further extend this Theorem to a necessary condition in Theorem. Define

$$f(\lambda_i) = -\rho a_i + \frac{\rho^2}{2}\lambda_i a_i + \frac{\rho^2}{2N_s}\sigma_i \lambda_i^3,$$

where $a_i := (\mathbf{v}_i^\top \mathbf{g})^2 > 0$ [1]. This is a cubic polynomial in $\lambda_i$ of the form

$$f(\lambda) = a + b\lambda + c\lambda^3,$$

with:

$$a = -\rho a_i < 0,$$

$$b = \frac{\rho^2}{2}a_i > 0,$$

$$c = \frac{\rho^2}{2N_s}\sigma_i > 0.$$

Since $f'(\lambda) = b + 3c\lambda^2 > 0$ for all $\lambda > 0$, the function is strictly increasing. Therefore, ensuring $f(\lambda_i) < 0$ is equivalent to requiring $\lambda_i$ to be smaller than the (unique) positive root of $f(\lambda) = 0$.

By Cardano's method for solving cubics, the condition $\Delta = \left(\frac{b}{3c}\right)^3 + \left(\frac{a}{2c}\right)^2 > 0$ implies a unique real root. The root can be expressed as:

$$\lambda_i = \left(\frac{z_i}{\rho}\right)^{1/3}\left(\left(1 + \sqrt{1 + \frac{z_i\rho^2}{27}}\right)^{1/3} + \left(1 - \sqrt{1 + \frac{z_i\rho^2}{27}}\right)^{1/3}\right) \tag{24}$$

$$= 2\sqrt{\frac{z_i}{3}}\sinh\left(\frac{1}{3}\sinh^{-1}\left(\frac{3}{\rho}\sqrt{\frac{3}{z_i}}\right)\right), \tag{25}$$

where $z_i := \frac{N_s a_i}{\sigma_i} = \frac{N_s(\mathbf{v}_i^\top \mathbf{g})^2}{\sigma_i}$.

Hence, the expected loss decreases if

$$0 < \lambda_i < 2\sqrt{\frac{z_i}{3}}\sinh\left(\frac{1}{3}\sinh^{-1}\left(\frac{3}{\rho}\sqrt{\frac{3}{z_i}}\right)\right) \quad \text{for all } i.$$

Now that we have established a sufficient condition on the curvature matrix $\mathbf{Q}$ to ensure that the expected loss decreases, we next address the random variability of the actual loss change due to the stochasticity of the update. Specifically, we show that the random variable $\Delta\ell$ concentrates sharply around its expectation.

For simplicity, we assumed to $\mathbf{Q}$ to be full rank. If $\mathbf{Q}$ has a non-trivial null-space that is $\lambda_i = 0$ for some $d > i \geq k$, then we have $\mathbf{v}_i^\top \mathbf{g} = \mathbf{v}_i^\top \mathbf{Q}\mathbf{m}_t = \lambda_i \mathbf{v}_i^\top \mathbf{m}_t = 0$. This means there is no component of the iterate in the null-space direction of $\mathbf{Q}$ and does not contribute to the descent objective.

This guarantees that, under the sufficient condition derived above, the actual loss decrease also holds with high probability, not just in expectation. The following lemma provides a concentration bound for $\Delta\ell$ about its mean:

**Lemma 3.2.** *In the same setting as Theorem 3.1, when the expected loss at next iteration is smaller than the previous loss by some margin $\delta > 0$, that is,*

$$\mathbb{E}_{\boldsymbol{\epsilon}}[\ell(\mathbf{m}_{t+1}^{\boldsymbol{\epsilon}})] < \ell(\mathbf{m}_t) - \delta,$$

*then $\ell(\mathbf{m}_{t+1}^{\boldsymbol{\epsilon}}) - \ell(\mathbf{m}_t) < 0$ occurs with probability at least $1 - 2\exp\left(-c_1 \min\left\{\delta^2 N_s^2/c_2, \delta N_s/c_2\right\}\right)$, for constants $c_1, c_2 > 0$ depending only on $\rho$, $\mathbf{Q}$, and $\mathbf{\Sigma}$.*

---

[1]For GD, if $\mathbf{v}_i^T \mathbf{m}_0 = 0$, $a_i = 0$ always holds. However, stochasticity ensures that $a_i \neq 0$ even when an $\mathbf{m}_0$ is chosen such that it is orthogonal to some $\mathbf{v}_i$'s.

*Proof.* In Theorem 3.1, we showed that the descent step occurs in *expectation*, if the sharpness is less than the stability threshold. In this lemma, we show that the descent step occurs with high probability under the same condition. To show, this we use derive a concentration inequality to show that the descent step concentrates around its expectation with high probability. Moreover, with larger MC samples $N_s$, the deviation is small. The proof occurs in the following steps:

1. Let $\epsilon = \hat{\mathbf{g}} - \mathbf{g}$ denote the gradient noise which is a rv $\epsilon \sim \mathcal{N}(\mathbf{0}, \frac{1}{N_s} \mathbf{Q} \mathbf{\Sigma} \mathbf{Q})$. The descent step has both linear and quadratic terms wrt $\epsilon$.

2. To ensure the deviation of the descent $\Delta\ell$ from its expectation $\mathbb{E}[\Delta\ell]$, we analyze the concentration of both the linear term and the quadratic term. The concentration of the linear term follows simply from the Gaussian tail. The concentration of the quadratic term is done using the Hanson Wright concentration inequality Rudelson and Vershynin [2013], Vershynin [2018].

3. The tail bounds from both the linear and the quadratic term is combined using the union bound which concludes the proof.

**Step-1**: *Separating fluctuation and expectation term*

The descent step $\Delta\ell$ as derived in Theorem 3.1 can be written as its expectation $\mathbb{E}[\Delta\ell]$ and fluctuation $R$.

$$\Delta\ell = l(\mathbf{m}_{t+1}) - l(\mathbf{m}_t) = -\rho\mathbf{g}^T\hat{\mathbf{g}} + \frac{\rho^2}{2}\hat{\mathbf{g}}^T\mathbf{Q}\hat{\mathbf{g}}$$

$$= -\rho\mathbf{g}^T(\mathbf{g}+\epsilon) + \frac{\rho^2}{2}(\mathbf{g}+\epsilon)^T\mathbf{Q}(\mathbf{g}+\epsilon)$$

$$= \underbrace{-\rho\|\mathbf{g}\|^2 + \frac{\rho^2}{2}\mathbf{g}^T\mathbf{Q}\mathbf{g} + \frac{\rho^2}{2}\mathbb{E}[\epsilon^T\mathbf{Q}\epsilon]}_{\mathbb{E}[\Delta\ell]} + \underbrace{(-\rho\mathbf{g}^T\epsilon + \rho^2\mathbf{g}^T\mathbf{Q}\epsilon + \frac{\rho^2}{2}(\epsilon^T\mathbf{Q}\epsilon - \mathbb{E}[\epsilon^T\mathbf{Q}\epsilon]))}_{R}$$

We show that the fluctuation $R = \Delta\ell - \mathbb{E}[\Delta\ell]$ is small with high probability, i.e, it has a sub-Gaussian tail.

**Step-2**: *Concentration bound on the fluctuation*

We separate the fluctuation on linear and quadratic terms wrt $\epsilon$ since applying concentration inequality on each term is different.

$R = L + Q$, where $L = -\rho\mathbf{g}^T\epsilon + \rho^2\mathbf{g}^T\mathbf{Q}\epsilon$ and $Q = \frac{\rho^2}{2}(\epsilon^T\mathbf{Q}\epsilon - \mathbb{E}[\epsilon^T\mathbf{Q}\epsilon])$.

Tail bound on L using sub-Gaussin concentration: Since $L$ is a linear function of the Gaussian vector $\epsilon$, it itself is Gaussian. its variance is

$$\sigma_L^2 = \frac{1}{N_s}\mathbf{h}^T(\mathbf{Q}\mathbf{\Sigma}\mathbf{Q})\mathbf{h} = \frac{\alpha_L}{N_s}$$

where $\alpha_L := \mathbf{h}^T(\mathbf{Q}\mathbf{\Sigma}\mathbf{Q})\mathbf{h}$ and $\mathbf{h} = \rho^2\mathbf{Q}\mathbf{g} - \rho\mathbf{g}$. A standard Gaussian tail then implies that for any $t > 0$,

$$Pr\big(|L| \geq t\big) \leq 2\exp\big(-\frac{t^2}{2\sigma_L^2}\big) = 2\exp\big(-\frac{t^2 N_s}{2\alpha_L}\big)$$

Choosing $t = \frac{\epsilon}{2}$ yields

$$Pr\big(|L| \geq \frac{\epsilon}{2}\big) \leq 2\exp\big(-\frac{\epsilon^2 N_s}{8\alpha_L}\big) \tag{26}$$

Tail bound on Q using Hanson-Wright Concentration inequality: For the quadratic term, note that $Q = \frac{\rho^2}{2}(\epsilon^T\mathbf{Q}\epsilon - \mathbb{E}[\epsilon^T\mathbf{Q}\epsilon])$. The Hanson-Wright inequality states that a sub-Gaussian vector $\epsilon$ (here Gaussian) with sub-Gaussian norm bounded by $K$ for any matrix $\mathbf{A}$ (in our case $\mathbf{A} = \frac{\rho^2}{2}\mathbf{Q}$), there

exists universal constant $c_1 > 0$ such that for any $t > 0$,

$$\Pr\left(\left|\boldsymbol{\epsilon}^T \mathbf{A}\boldsymbol{\epsilon} - \mathbb{E}[\boldsymbol{\epsilon}^T \mathbf{A}\boldsymbol{\epsilon}]\right| > t\right) \le 2\exp\left(-c_1 \min\left\{\frac{t^2}{K^4\|\mathbf{A}\|_F^2}, \frac{t}{K^2\|\mathbf{A}\|_2}\right\}\right).$$

Since $\boldsymbol{\epsilon} \sim \mathcal{N}(\mathbf{0}, \frac{1}{N_s}\mathbf{Q}\boldsymbol{\Sigma}\mathbf{Q})$, its sub-Gaussian norm satisfies $K^2 \le c_2^2 \lambda_{max}\left(\frac{1}{N_s}\mathbf{Q}\boldsymbol{\Sigma}\mathbf{Q}\right) = \frac{c_2^2 \beta}{N_s}$, where we define $\beta = \lambda_{max}(\mathbf{Q}\boldsymbol{\Sigma}\mathbf{Q})$ and $c_2$ is another universal constant. Furthermore, we have $K^4 < \frac{c_4^2 \beta^2}{N_s^2}$, $\|\mathbf{A}\| = \frac{\rho^2}{2}\|\mathbf{Q}\|$ and $\|\mathbf{A}\|_F = \frac{\rho^2}{2}\|\mathbf{Q}\|_F$. To substitute the sub-gaussian norm $K$ from the bound, we use the inequalities $\frac{t^2}{K^4\|\mathbf{A}\|_F^2} \ge \frac{t^2 N_s^2}{c_2^4 \beta^2 \frac{\rho^4}{4}\|\mathbf{Q}\|_F^2}$ and $\frac{t}{K^2\|\mathbf{A}\|} \ge \frac{t N_s}{c_2^2 \beta \frac{\rho^2}{2}\|\mathbf{Q}\|}$, we get the tail bound to be

$$\Pr\left(|Q| > t\right) \le 2\exp\left(-c_1 \min\left\{\frac{t^2 N_s^2}{c_2^4 \beta^2 \frac{\rho^4}{4}\|\mathbf{Q}\|_F^2}, \frac{t N_s}{c_2^2 \beta \frac{\rho^2}{2}\|\mathbf{Q}\|}\right\}\right).$$

Finally chosing $c_3 = \max\{c_2^4 \beta^2 \frac{\rho^4}{4}\|\mathbf{Q}\|_F^2, c_2^2 \beta \frac{\rho^2}{2}\|\mathbf{Q}\|\}$ and $t = \frac{\epsilon}{2}$, we get:

$$\Pr\left(|Q| > \frac{\epsilon}{2}\right) \le 2\exp\left(-c_1 \min\left\{\frac{(\frac{\epsilon}{2})^2 N_s^2}{c_3}, \frac{(\frac{\epsilon}{2})N_s}{c_3}\right\}\right). \tag{27}$$

**Step-3**: *Combining L and Q concentration using union bound*

Since $R = L + Q$, by the union bound and using equation (27) and (26) we get

$$\Pr\left(|R| > \epsilon\right) \le \Pr\left(|L| > \frac{\epsilon}{2}\right) + \Pr\left(|Q| > \frac{\epsilon}{2}\right)$$

$$\implies \Pr\left(|R| > \epsilon\right) \le 2\exp\left(-\frac{\epsilon^2 N_s}{8\alpha_L}\right) + 2\exp\left(-c_1 \min\left\{\frac{(\frac{\epsilon}{2})^2 N_s^2}{c_3}, \frac{(\frac{\epsilon}{2})N_s}{c_3}\right\}\right)$$

We combine the sum to a single exponential tail by using the inequality $\exp(-X) + \exp(-Y) \le 2\exp(-\min\{X,Y\})$. Now chosing $d_2 = 32\max\{\alpha_L, c_3\} = 32\max\{\alpha_L, c_2^4 \beta^2 \frac{\rho^4}{4}\|\mathbf{Q}\|_F^2, c_2^2 \beta \frac{\rho^2}{2}\|\mathbf{Q}\|\}$ (substituting $c_3$) and $d_1 = \min\{\frac{1}{32\alpha_L}, \frac{c_1}{4}\}$ and using $\exp(-X) + \exp(-Y) \le 2\exp(-\min\{X,Y\})$, we get

$$\Pr\left(|R| > \epsilon\right) \le 2\exp\left(-d_1 \min\left\{\frac{\epsilon^2 N_s^2}{d_2}, \frac{\epsilon N_s}{d_2}\right\}\right)$$

Assume the *expected* descent step is strictly negative by some margin $\delta > 0$, i.e. $\mathbb{E}[\Delta\ell] \le -\delta < 0$. Recall that $R = \Delta\ell - \mathbb{E}[\Delta\ell]$. Under this assumption, if $\Delta\ell \ge 0$, then $\Delta\ell - \mathbb{E}[\Delta\ell] \ge \delta$. Hence

$$\{\Delta\ell \ge 0\} \subseteq \{|R| \ge \delta\}.$$

We set $\epsilon = \delta$ in the concentration bound

$$\Pr(|R| \ge \epsilon) \le 2\exp\left(-d_1 \min\left\{\frac{\epsilon^2 N_s^2}{d_2}, \frac{\epsilon N_s}{d_2}\right\}\right),$$

and obtain

$$\Pr(\Delta\ell \ge 0) \le 2\exp\left(-d_1 \min\left\{\frac{\delta^2 N_s^2}{d_2}, \frac{\delta N_s}{d_2}\right\}\right).$$

Therefore, with probability at least

$$1 - 2\exp\left(-d_1 \min\left\{\frac{\delta^2 N_s^2}{d_2}, \frac{\delta N_s}{d_2}\right\}\right),$$

we have $\Delta\ell < 0$. Thus, if the expected descent step is at most $-\delta$, then $\Delta\ell$ is negative with high probability. $\qquad\square$

## C  Role of posterior samples on a quadratic

On a quadratic loss, the perturbed averaged gradient follows a Gaussian distribution with variance inversely proportional to the number of posterior samples:

$$\hat{g} \sim \mathcal{N}\left(\nabla\ell(\boldsymbol{m}_t), \frac{1}{N_s}\mathbf{Q}\boldsymbol{\Sigma}\mathbf{Q}\right). \tag{28}$$

For large $N_s$, the dynamics of variational GD approach those of standard GD and exhibit similar stability characteristics. To examine the role of $N_s$, we consider a quadratic loss $\ell(m) = \frac{a}{2}m^2$ and compare GD with variational GD across varying values of $N_s$. When $\rho < 2/a$, standard GD converges to the minimum. In the limit $N_s \to \infty$, variational GD recovers this behavior. However, for finite $N_s$, the gradient estimate becomes noisier, reducing the stability threshold as predicted by Theorem-1. Figure11 shows the resulting trajectories and histograms of the iterates. As $N_s$ decreases, the iterates exhibit greater variability and cover a wider range, reflecting increased instability. These results confirm the theoretical prediction that smaller $N_s$ increases the likelihood of the loss increasing in the next step, even under a stable learning rate.

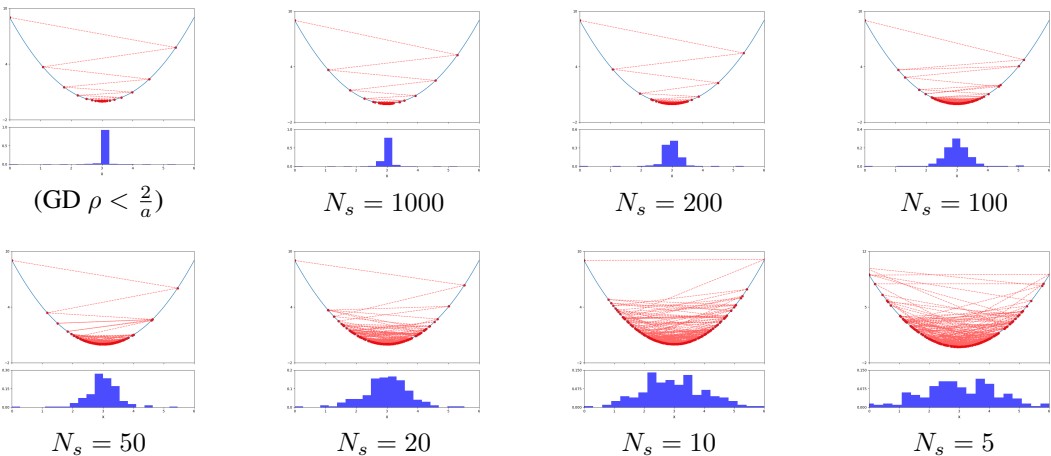

Figure 11: Histogram plot of iterates on a quadratic for GD with noise injection from a posterior distribution with $N_s$ samples. As $N_s$ decreases, the iterates become more unstable.

We further visualize the stability threshold predicted by Theorem-1 in Figure12. For each posterior sample size $N_s$, we compute the descent probability on a quadratic loss and plot it alongside the corresponding stability threshold $2/\rho \cdot \mathrm{VF}$. As shown in the top row, descent occurs with high probability whenever the curvature is below the threshold—consistent with Theorem 1. In the bottom row, we binarize the descent probability by setting it to 1 if it exceeds 0.5, and 0 otherwise. The resulting transition boundary closely aligns with the predicted threshold, further validating our theoretical result.

## D  Loss Smoothing Beyond Local Quadratic Approximation

In our paper, we consider a local quadratic approximation of the loss to characterize the distribution of the perturbed gradient. By the property that a Gaussian distribution is invariant under linear transformation, we showed that the perturbed gradient (or it's sample average) is also Gaussian. This is observed by characterizing the distribution of the perturbed gradient $\nabla\ell(\mathbf{m}_t + \boldsymbol{\epsilon})$. Employing a Taylor expansion around the current mean parameter $\mathbf{m}$, we obtain:

$$\nabla\ell(\boldsymbol{m} + \boldsymbol{\epsilon}) = \nabla\ell(\boldsymbol{m}) + \nabla^2\ell(\boldsymbol{m})\boldsymbol{\epsilon} + \frac{1}{2}\nabla^3\ell(\boldsymbol{m})[\boldsymbol{\epsilon}, \boldsymbol{\epsilon}] + O(\|\boldsymbol{\epsilon}\|^3), \quad \boldsymbol{\epsilon} \sim \mathcal{N}(\mathbf{0}, \boldsymbol{\Sigma}) \tag{29}$$

In general, the distribution of $\nabla\ell(\boldsymbol{m} + \boldsymbol{\epsilon})$ involves higher-order terms of Gaussian variables, making exact characterization challenging. However, if the perturbation covariance $\boldsymbol{\Sigma}$ is sufficiently small relative to the local curvature, defined by the Hessian $\mathbf{H}(\boldsymbol{m}) = \nabla^2\ell(\boldsymbol{m})$, specifically, when $\|\nabla^3\ell(\boldsymbol{m})\|_{op} \cdot \|\boldsymbol{\Sigma}\|_2 \ll \|\mathbf{H}(\boldsymbol{m})\|_2$, the gradient approximation effectively simplifies to a linear approximation $\nabla\ell(\boldsymbol{m} + \boldsymbol{\epsilon}) \approx \nabla\ell(\boldsymbol{m}) + \mathbf{H}(\boldsymbol{m})\boldsymbol{\epsilon}$. This approximation is exact

for a quadratic. Under this condition, since $\epsilon \sim \mathcal{N}(\mathbf{0}, \boldsymbol{\Sigma})$, the perturbed gradient is Gaussian $\nabla\ell(\boldsymbol{m} + \epsilon) \sim \mathcal{N}(\nabla\ell(\boldsymbol{m}), \mathbf{H}(\boldsymbol{m})\boldsymbol{\Sigma}\mathbf{H}(\boldsymbol{m})^{\top})$.

However, in high-perturbation regimes (large covariance $\Sigma$ where $\|\nabla^3\ell(\boldsymbol{m})\|_{op} \cdot \|\boldsymbol{\Sigma}\|_2 \approx \|\mathbf{H}(\boldsymbol{m})\|_2$), the perturbed gradient is no more Gaussian (since it involves second-order terms on $\epsilon$). Furthermore, the expectation of the perturbed gradient has a contribution from the neighborhood of the loss, which is referred to here as *third-order curvature bias*.

$$\mathbb{E}[\nabla\ell(\mathbf{m} + \epsilon)] \approx \nabla\ell(\mathbf{m}) + \frac{1}{2}\underbrace{\mathrm{Tr}_{2,3}(\boldsymbol{\Sigma}\nabla^3\ell(\mathbf{m}))}_{\text{third-order curvature bias}} + \mathcal{O}(\|\boldsymbol{\Sigma}\|^2). \tag{30}$$

Having a large number of posterior samples, can help recover this modified expectation better with increasing number of samples. Here, the algorithm effectively follows the gradient of a smoothed version of the loss landscape, since samples are drawn from a wide neighborhood around $\boldsymbol{m}$. Accurately estimating this biased but meaningful descent direction under heavy-tailed noise requires a sufficiently large $N_s$, not just to reduce variance, but to faithfully recover the expected smoothed gradient.

In the next section and the subsequent theorem, we study this phenomenon in a non-quadratic function. Here we show that, first for a quadratic function, smoothing does not change the curvature of the underlying loss. But for a quartic function, smoothing by expectation changes the underlying curvature by the variance of the posterior, and furthermore, for smoothing by finite averaging, the curvature changes as a function of both the variance and the number of samples used to approximate the expectation.

Let $\ell(\theta) = a\theta^2 + b\theta + c$ and let $q = \mathcal{N}(\theta|m, \sigma^2)$, then

$$\ell_{conv}(m) = \mathbb{E}_q\ell(\theta) = \mathbb{E}_q(a\theta^2 + b\theta + c) = am^2 + bm + c + a\sigma^2$$

So w.r.t the reparameterized variable $m$, the curvature of the loss remains unchanged, only shift occurs, proportional to $\sigma^2$. So, the stability dynamics on $\ell(\theta)$ and $\ell_{conv}(m)$ are the same. The loss diverges only when the learning rate is $\rho > \frac{2}{a}$. However, this is not the case with general losses, especially for losses where $\ell^{(4)}(\theta) \neq 0$, i.e if it has a non-zero fourth order derivative.

For example, let's take the example of a quartic function $\ell(\theta) = (\theta^2 - 1)^2$ where minima is at $\theta^* = \pm 1$ and $\ell''(\theta^*) = 12\theta^{*2} - 4 = 8$. Now, the smoothed loss $\ell_{conv}(m) = \mathbb{E}_q\ell(\theta) = \mathbb{E}_q(\theta^2 - 1)^2 = m^4 + m^2(6\sigma^2 - 2) + (3\sigma^4 - 2\sigma^2 + 1)$. The new loss has a global minima at $m^* = \pm\sqrt{1 - 3\sigma^2}$

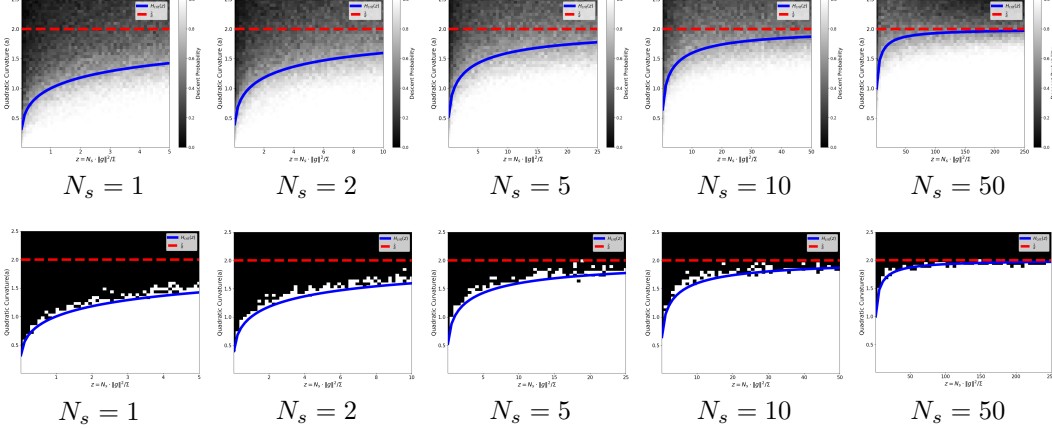

Figure 12: (Top Row): Probability of descent evaluated over 10 trials on several simulations of combination of quadratic curvature and noise variance. (Bottom Row): Hard thresholded probability for descent. The transition occurs near the boundary given by the derived expression of $2/\rho \cdot \mathrm{VF}$ from Theorem-1.

(only if $\sigma < \sqrt{\frac{1}{3}}$). Comparing the curvature of both the losses at the global minima, we have:

$$\ell''(\theta) = 12\theta^2 - 4 = 8$$
$$\ell''_{conv}(m) = 12m^2 + 2(6\sigma^2 - 2) = 12m^2 - 4 + 12\sigma^2 = 8 - 24\sigma^2$$

So, the smoothing operator, changes the curvature at the global minima for quartic functions. note that this wasn't the case for quadratic functions, where the curvature was unchanged. This further changes the stability dynamics since the learning rate required for stable convergence is difference, for original loss $\ell(\theta)$, it is $\rho < \frac{2}{8}$, whereas for the smoothed loss, it is $\rho < \frac{2}{8-24\sigma^2}$. However, consider the effect of averaging of loss using finite mc samples $N_s$, $\ell_{avg}(\theta) = \frac{1}{N_s} \sum_{i=1}^{N_s} \ell(\theta + \epsilon_i)$, where $\epsilon \sim \mathcal{N}(0, \sigma^2)$:

$$\ell_{avg}(\theta) = \frac{1}{N_s} \sum_{i=1}^{N_s} \ell(\theta + \epsilon_i)$$

$$= \frac{1}{N_s} \sum_{i=1}^{N_s} ((\theta + \epsilon_i)^2 - 1)^2$$

$$= \theta^4 + \theta^2 \left( \frac{1}{N_s} \sum_i^{N_s} (6\epsilon_i^2 + 4\epsilon_i - 2) \right) + 4\theta \left( \frac{1}{N_s} \sum_i^{N_s} \epsilon_i(\epsilon_i^2 - 1) \right) + \left( \frac{1}{N_s} \sum_i^{N_s} (\epsilon_i^4 - 2\epsilon_i^2 + 1) \right)$$

So, the second derivative becomes:

$$\ell''_{avg}(\theta) = 12\theta^2 - 4 + \frac{2}{N_s} \sum_i^{N_s} (6\epsilon_i^2 + 4\epsilon_i)$$

If we compare the second derivative of the smoothed out-loss $\ell''_{conv}(m)$ and averaged loss over $N_s$ samples $\ell''_{avg}(\theta)$, we observe that wrt samples, the expectation is same, i.e,

$$\mathbb{E}_{\epsilon_i} \ell''_{avg}(\theta) = 12\theta^2 - 4 + \mathbb{E}_{\epsilon_i} [\frac{2}{N_s} \sum_i^{N_s} (6\epsilon_i^2 + 4\epsilon_i)] = 12\theta^2 - 4 + 12\sigma^2 = \ell''_{conv}(\theta)$$

However, the pointwise deviation of $\ell''_{avg}(\theta)$ and $\ell''_{conv}(\theta)$ depends on the number of samples $N_s$ Taking the difference we have:

$$\ell''_{avg}(\theta) - \ell''_{conv}(\theta) = \frac{2}{N_s} \sum_{i=1}^{N_s} \left[ (6\epsilon_i^2 + 4\epsilon_i) - 6\sigma^2 \right].$$

Define
$$Y_i = (6\epsilon_i^2 + 4\epsilon_i) - 6\sigma^2.$$
Since $\mathbb{E}[\epsilon_i^2] = \sigma^2$ and $\mathbb{E}[\epsilon_i] = 0$, each $Y_i$ has mean zero:
$$\mathbb{E}[Y_i] = 0.$$
Hence,

$$\ell''_{avg}(\theta) - \ell''_{conv}(\theta) = \frac{2}{N_s} \sum_{i=1}^{N_s} Y_i.$$

Note that $\epsilon_i \sim \mathcal{N}(0, \sigma^2)$ is sub-Gaussian, and $\epsilon_i^2$ is sub-exponential. Thus $Y_i = 6\epsilon_i^2 + 4\epsilon_i$ is a linear combination of a sub-exponential and a sub-Gaussian variable, which remains *sub-exponential*. Shifting by the constant $-6\sigma^2$ does not affect sub-exponential parameters. Therefore, each $Y_i$ is sub-exponential.

Let $(v, b)$ be sub-exponential parameters for $Y_i$. By standard Bernstein-type concentration for sub-exponential random variables, there exists a universal constant $c > 0$ such that for all $t > 0$:

$$\Pr\left( \left| \frac{1}{N_s} \sum_{i=1}^{N_s} Y_i \right| \geq t \right) \leq 2 \exp\left( -c N_s \min\left\{ \frac{t^2}{v}, \frac{t}{b} \right\} \right).$$

Since

$$\ell_{avg}^{''}(\theta) - \ell_{conv}^{''}(\theta) \;=\; \frac{2}{N_s} \sum_{i=1}^{N_s} Y_i,$$

we have

$$\left| \ell_{avg}^{''}(\theta) - \ell_{conv}^{''}(\theta) \right| \;=\; \frac{2}{N_s} \left| \sum_{i=1}^{N_s} Y_i \right| \;=\; 2 \left| \tfrac{1}{N_s} \sum_{i=1}^{N_s} Y_i \right|.$$

Hence, for any $\delta > 0$,

$$\Pr\!\Big( \left| \ell_{avg}^{''}(\theta) - \ell_{conv}^{''}(\theta) \right| \geq \delta \Big) \;=\; \Pr\!\Big( \left| \tfrac{1}{N_s} \sum_{i=1}^{N_s} Y_i \right| \geq \tfrac{\delta}{2} \Big) \;\leq\; 2 \exp\!\Big( -c\, N_s\, \min\Big\{ \tfrac{(\delta/2)^2}{v}, \tfrac{\delta/2}{b} \Big\} \Big).$$

Plugging in $v = c_1 \sigma^4$ and $b = c_2 \sigma^2$, which are the subexponential norms in terms of the variance, we get for some constant $c_1$ and $c_2$:

$$\Pr\!\Big( \left| \ell_{avg}^{''}(\theta) - \ell_{conv}^{''}(\theta) \right| \geq \delta \Big) \;\leq\; 2 \exp\!\Big( -c\, N_s\, \min\Big\{ \tfrac{\delta^2}{4 c_1 \sigma^4}, \tfrac{\delta}{2 c_2 \sigma^2} \Big\} \Big).$$

This inequality shows that the finite-sample second derivative $\ell_{avg}^{''}(\theta)$ concentrates around the infinite-sample second derivative $\ell_{conv}^{''}(\theta)$ at an *exponential* rate in $N_s$. While they are identical in expectation, the above result quantifies their pointwise deviation with high probability. We formalize this observation for general analytical function which has continous derivatives.

**Theorem D.1** (Concentration of Smoothed Curvature). *Let $f : \mathbb{R} \to \mathbb{R}$ be six-times contin-uously differentiable analytical function, with all derivatives up to order 6 bounded. Suppose $\sigma^2 \sup_x f^{(6)}(x) < \sup_x f^{(4)}(x)$ and for i.i.d. samples $\{\epsilon_i\}_{i=1}^{N_s}$ with $\epsilon_i \sim \mathcal{N}(0, \sigma^2)$, define*

$$f_{\text{avg}}(x) \;=\; \frac{1}{N_s} \sum_{i=1}^{N_s} f\big( x + \epsilon_i \big), \quad f_{\text{conv}}(x) \;=\; \mathbb{E}_{z \sim \mathcal{N}(0, \sigma^2)}\big[ f(x + z) \big].$$

*Then there exist universal constants $c, c_1, c_2 > 0$ such that for any $\delta > 0$,*

$$\Pr\!\Big( \left| f_{\text{avg}}''(x) - f_{\text{conv}}''(x) \right| \geq \delta \Big) \;\leq\; 2 \exp\!\Big( -c\, N_s\, \min\Big\{ \tfrac{\delta^2}{4 c_1 \sigma^4}, \tfrac{\delta}{2 c_2 \sigma^2} \Big\} \Big).$$

*Hence, $\left| f_{\text{avg}}''(x) - f_{\text{conv}}''(x) \right|$ concentrates around zero at an exponential rate in $N_s$.*

A general extension of this proof can be done for functions which has a finite fourth order derivative and small sixth order derivative such that $\sigma^2 \sup_x f^6(x) < \sup_x f^4(x)$. Starting with a sufficiently smooth function $f \in C^k$, we can expand $f(x + z)$ in a Taylor series around $x$. Let $z \sim \mathcal{N}(0, \sigma^2)$. Then:

$$f(x + z) \;=\; f(x) \;+\; z\, f'(x) \;+\; \frac{z^2}{2!}\, f''(x) \;+\; \frac{z^3}{3!}\, f^{(3)}(x) \;+\; \cdots$$

Since $z$ is Gaussian with zero mean, all the *odd* moments $\mathbb{E}[z]$, $\mathbb{E}[z^3]$, etc., vanish. Thus, when we take the expectation,

$$f_{\text{conv}}(x) \;=\; \mathbb{E}_{z \sim \mathcal{N}(0, \sigma^2)}\big[ f(x + z) \big] \;=\; f(x) \;+\; \frac{\sigma^2}{2}\, f''(x) +\; +\; \frac{\sigma^4}{8}\, f^{(4)}(x) \;+\; \frac{\sigma^6}{48}\, f^{(6)}(x) \;+\; \cdots$$

In the above, we use the fact that $\mathbb{E}[z^2] = \sigma^2$, $\mathbb{E}[z^4] = 3\,\sigma^4$, $\mathbb{E}[z^6] = 15\,\sigma^6$, etc., and only the *even* powers contribute. The curvature of the original function then becomes:

$$f_{\text{conv}}^{''}(x) = f''(x) + \frac{\sigma^2}{2}\, f^4(x) + \frac{\sigma^4}{8}\, f^6(x) + ..$$

Under the condition that $\sigma^2 \sup_x f^6(x) < \sup_x f^4(x)$, we get

$$f_{\text{conv}}^{''}(x) \approx f''(x) + \frac{\sigma^2}{2}\, f^4(x) \tag{31}$$

The smoothing effect does change the second order curvature of the loss. Furthermore, approximating This approximation makes the noise due to the $N_s$ samples be a subexponential and similar result holds.

# E   Discussion on stability and descent for Variational GD

*On Necessity and Sufficiency of descent:*

A further critical difference lies in the nature of the descent conditions for GD versus VGD. For GD on a quadratic loss, the standard descent lemma provides a condition on the maximum eigenvalue $\lambda_{max} < 2/\rho$ that is both necessary and sufficient for ensuring stability and monotonic descent. A violation of this single condition guarantees divergence.

For VGD, the analysis is more nuanced. The true necessary and sufficient condition for the expected loss to decrease is that the sum of all components in its eigen-decomposition must be negative, as shown below:

$$\mathbb{E}[\Delta\ell] = -\rho\mathbf{g}^\top(\mathbf{I} - \frac{\rho}{2}\mathbf{Q})\mathbf{g} + \frac{\rho^2}{2N_s}\operatorname{Tr}\left(\mathbf{\Sigma}\mathbf{Q}^3\right) = \sum_{i=1}^{d} f(\lambda_i, \mathbf{v}_i) < 0 \tag{32}$$

However, analyzing this sum can be intractable. Instead, our work derives a practical sufficient condition by requiring each term in the sum to be negative independently, i.e., $f(\lambda_i, \mathbf{v}_i) < 0$ for all i. This is the condition presented in Theorem 1, which yields the stability limit $\lambda_i < 2/\rho \cdot \operatorname{VF}(z_i)$ for each eigenvalue.

This theoretical framework is validated by our extensive experiments on MLPs and ResNets across various learning-rate and variance. We consistently find that the leading Hessian eigenvalues, $\lambda_i$, hover around their respective stability thresholds, $2/\rho \cdot \operatorname{VF}(z_i)$. This alignment provides strong empirical support for our sufficient condition, showing it is an active constraint that accurately describes the behavior of VGD in practice.

Mulayoff and Michaeli [2024] also derives a stability condition (see their Theorem 5) which is a necessary and sufficient condition for the stability of SGD, but their setting is fundamentally different from ours. Firstly, in their setting, the assumption is that the batch-size is drawn uniformly at random from the finite set of all possible data samples. Our work, in contrast, does not model noise from data sampling but rather from perturbations to the model's weights, which we assume follow an explicit, continuous distribution (e.g., Gaussian). This leads to fundamentally different noise structures. In our VGD framework, the resulting gradient estimator is shown to follow a normal distribution whose covariance, $\frac{1}{N_s}\mathbf{Q}\mathbf{\Sigma}\mathbf{Q}$, is shaped by the Hessian ($\mathbf{Q}$), meaning noise is amplified in directions of high curvature. In the SGD paper, the gradient noise has a discrete distribution with a covariance determined by the dataset's intrinsic variance.

# F   Additional experiments across diverse datasets

While in the manuscript, we presented experiments on CIFAR-10, here we present several additional results to support our theorem and claims.

*Gaussian Variational GD:* In Figure 13, we compare GD and GD with weight perturbation trained across three different architectures on SVHN dataset Netzer et al. [2011]. We arrive at a similar conclusion that with Gaussian weight perturbation achieves smaller sharpness and better test accuracy than just GD. Similar trend is also observed for FashionMNIST dataset in Figure 15. In Figure 16, we show that in deep neural networks, sharpness also depends on the number of posterior samples, as smaller samples lead to smaller sharpness. In Figure 14, we plot the Variational factor along with the sharpness in ResNet and show that they match closely.

*Sharpness comparison for Cross Entropy Loss:* For cross-entropy (CE) loss, the sharpness dynamics differs from those observed with mean-squared error (MSE) loss. As training progresses and model predictions become increasingly confident, the term $p_i(1-p_i)$ in the Hessian approaches zero, driving the curvature and hence sharpness to vanish. This causes both GD and VL (or IVON) to converge to regions with negligible sharpness, although their transient behaviors differ. VL's sharper stability bound results in smaller peak sharpness compared to GD.

Figure 17 illustrates this effect for a fully connected neural network trained on CIFAR-10 with CE loss. The left panel shows the training loss, while the right panel presents the evolution of sharpness over training steps. Although both methods eventually exhibit vanishing curvature, IVON achieves

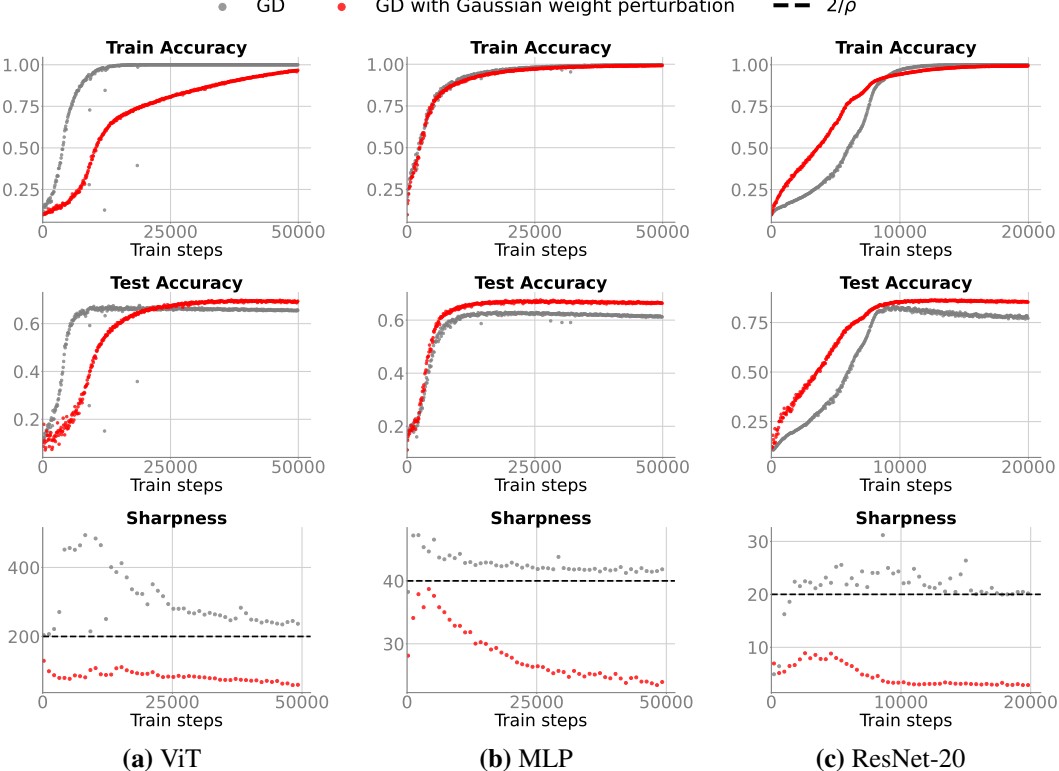

**(a)** ViT  **(b)** MLP  **(c)** ResNet-20

Figure 13: Similar to the trend in Fig. 4, weight perturbed GD consistently finds flatter minima on the SVHN dataset.

a substantially lower peak sharpness than GD. This highlights the improved stability and flatter convergence landscape induced by variational learning even under the CE loss.

*Weight-perturbation from heavy-tail distribution:* In the manuscript, we presented results on perturbation from a heavy-tailed Student-t distribution for MLP trained on CIFAR-10. In Figure 18, we plot the training dynamics across ViT and ResNet-20 models. Here, we observe that heavier tails (smaller $\alpha$) leads to smaller sharpness and better test accuracy.

*Experiments in Vision Transformers:* In Attention-based architectures, such as Vision Transformers, it has been widely observed that sharpness for GD often goes above the stability threshold $2/\rho$. This phenomenon has been widely studied [Zhai et al., 2023] as attention entropy collapse that makes training unstable in Vision Transformers, with sharp spikes in both the training and test accuracy. To mitigate such unstable behaviour, weight perturbation can be used, due to its sharpness reducing effect and thereby stabilizing training. For example, in Figure 20, we perform an ablation study of training ViTs with different perturbation covariance. Noise with larger variance consistently leads to more stable training and smaller sharpness in ViT. Similarly, for VON we observe that preconditioned sharpness is smaller than $2/\rho$ for ViT training 19.

*Experiments in NLP tasks:* To verify that Variational GD with isotropic Gaussian noise achieves lower sharpness compared to GD on an NLP task, we add a classification head to a frozen BERT-mini backbone and finetune the head on SST-2. For both GD and Variational GD, we use the same learning rate of 0.05. We report train loss, validation loss, and sharpness in Figure 21 and observe lower sharpness for Variational GD throughout. □

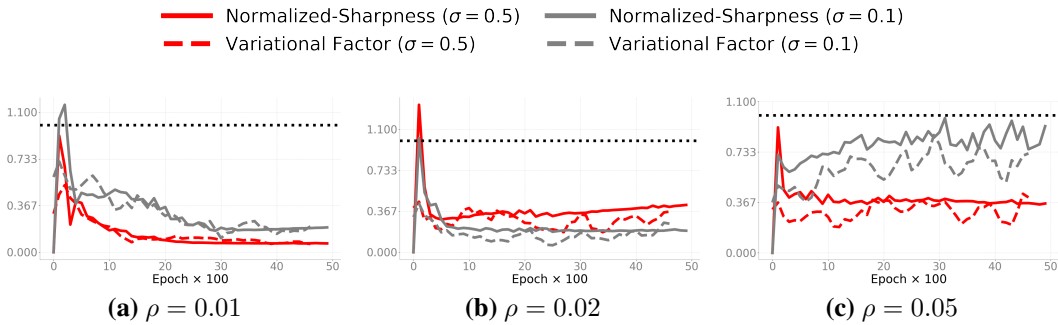

Figure 14: Normalized Sharpness $\|\nabla^2 \ell(\mathbf{m}_t)\|_2 / (2/\rho)$ hovers about the Variational Factor in ResNet.

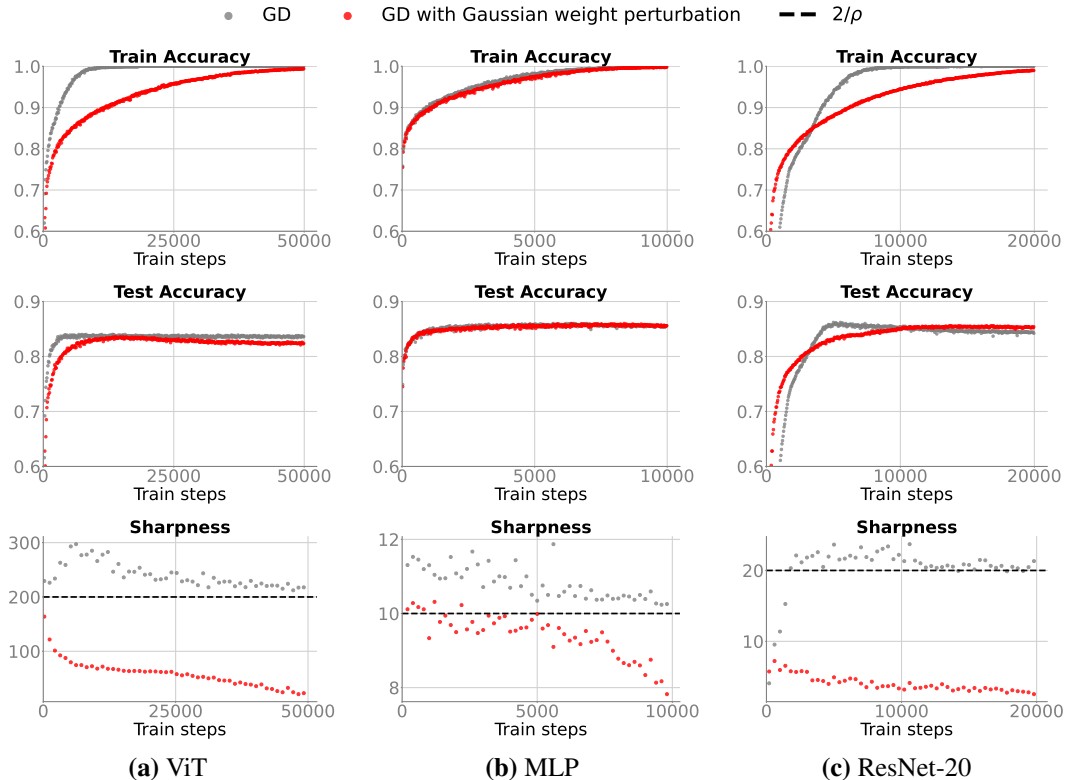

Figure 15: Similar to the trend in Fig. 4, variational learning finds flatter minima on the FashionM-NIST dataset.

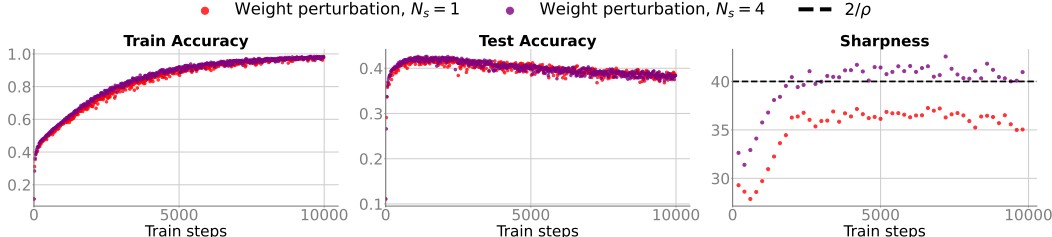

Figure 16: Using more samples per iteration leads to a large sharpness, as demonstrated in our ablation study where an MLP network is trained on a subset of the CIFAR-10 dataset containing 10000 images.

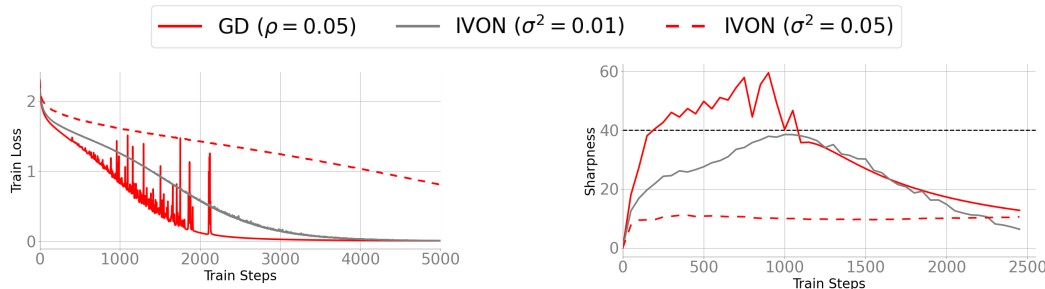

Figure 17: Training dynamics of fully connected neural network trained with GD and IVON with fixed covariance. Although the sharpness always drops to zero fro CE loss, IVON achives smaller peak sharpness than GD.

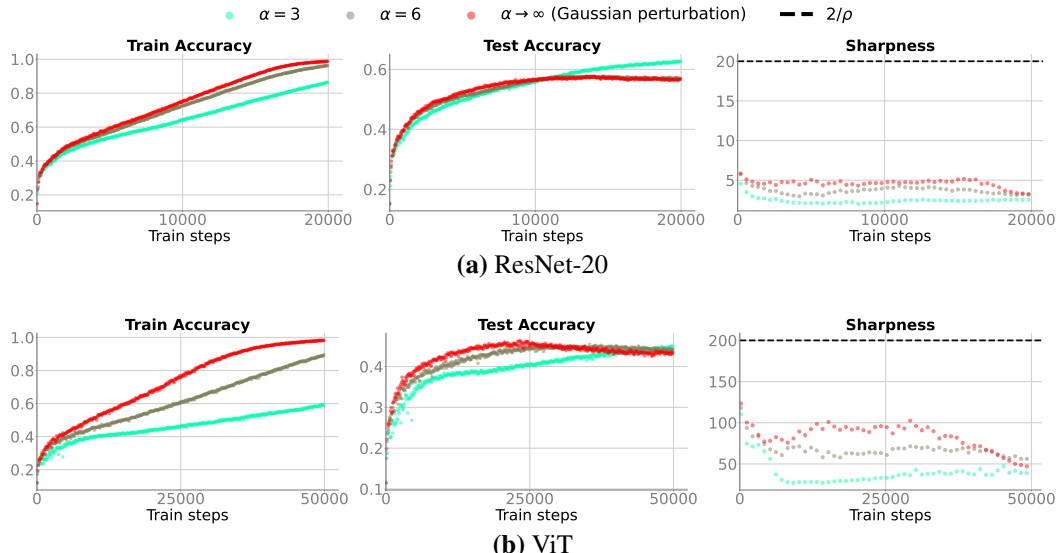

Figure 18: We notice similar trends shown in Fig. 7 in training ViT and ResNet models. That is to say, smaller $\alpha$ corresponds to heavier-tailed posterior which leads to smaller sharpness. Note that as $\alpha$ approaches infinity, the Gaussian posterior is recovered.

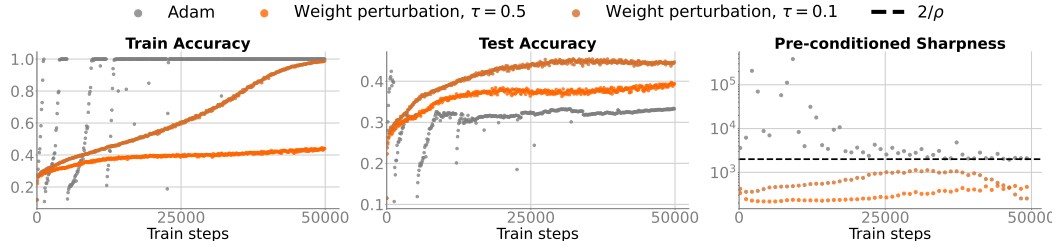

Figure 19: We notice similar trends shown in Fig. 8 in training ViT models. That is to say, Smaller temperatures which shrinks the posterior reaches larger preconditioned sharpness.

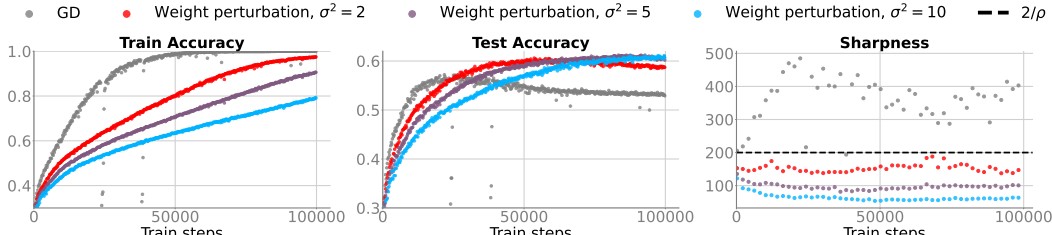

Figure 20: For ViT, GD training is prone to loss spikes, and the sharpness often goes above the stability threshold $2/\rho$. On the other hand, with weight perturbation the training becomes more stable, and larger noise variance consistently leads to smaller sharpness.

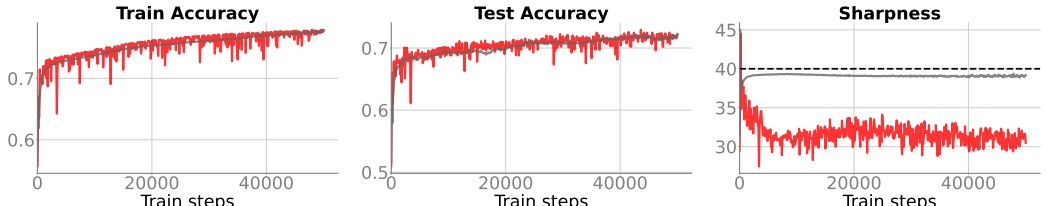

Figure 21: SST-2 head finetuning on a frozen BERT mini backbone comparing Variational GD with isotropic Gaussian noise to vanilla GD, both with learning rate 0.05.

Figure 22: Eigenspectrum ($\lambda_1$-$\lambda_{10}$) vs corresponding stability threshold $2/\rho \cdot \text{VF}(z_i)$. 2-layer MLP trained with VGD $\rho = 0.05$ and $\sigma^2 = 0.1$.

(a) $\lambda_1$      (b) $\lambda_2$

(c) $\lambda_3$      (d) $\lambda_4$

(e) $\lambda_5$      (f) $\lambda_6$

(g) $\lambda_7$      (h) $\lambda_8$

(i) $\lambda_9$      (j) $\lambda_{10}$

Figure 23: Eigenspectrum ($\lambda_1$-$\lambda_{10}$) vs corresponding stability threshold $2/\rho \cdot \text{VF}(z_i)$. 2-layer MLP trained with VGD $\rho = 0.02$ and $\sigma^2 = 0.5$.

Figure 24: Eigenspectrum ($\lambda_1$-$\lambda_{10}$) vs corresponding stability threshold $2/\rho \cdot \mathrm{VF}(z_i)$. 2-layer MLP trained with VGD $\rho = 0.02$ and $\sigma^2 = 1.0$.

Figure 25: Eigenspectrum ($\lambda_1$-$\lambda_{10}$) vs corresponding stability threshold $2/\rho \cdot \mathrm{VF}(z_i)$. 2-layer MLP trained with VGD $\rho = 0.1$ and $\sigma^2 = 1.0$.

Figure 26: Eigenspectrum ($\lambda_1$-$\lambda_{10}$) vs corresponding stability threshold $2/\rho \cdot \text{VF}(z_i)$. ResNet-20 trained with VGD $\rho = 0.1$ and $\sigma^2 = 0.1$.

(a) $\lambda_1$      (b) $\lambda_2$

(c) $\lambda_3$      (d) $\lambda_4$

(e) $\lambda_5$      (f) $\lambda_6$

(g) $\lambda_7$      (h) $\lambda_8$

(i) $\lambda_9$      (j) $\lambda_{10}$

Figure 27: Eigenspectrum ($\lambda_1$-$\lambda_{10}$) vs corresponding stability threshold $2/\rho \cdot \mathrm{VF}(z_i)$. ResNet-20 trained with VGD $\rho = 0.1$ and $\sigma^2 = 0.5$ (low variance).

(a) $\lambda_1$

(b) $\lambda_2$

(c) $\lambda_3$

(d) $\lambda_4$

(e) $\lambda_5$

(f) $\lambda_6$

(g) $\lambda_7$

(h) $\lambda_8$

(i) $\lambda_9$

(j) $\lambda_{10}$

Figure 28: Eigenspectrum ($\lambda_1$-$\lambda_{10}$) vs corresponding stability threshold $2/\rho \cdot \text{VF}(z_i)$. ResNet-20 trained with VGD $\rho = 0.05$ and $\sigma^2 = 0.1$.

Figure 29: Eigenspectrum $(\lambda_1$-$\lambda_{10})$ vs corresponding stability threshold $2/\rho \cdot \mathrm{VF}(z_i)$. ResNet-20 trained with VGD $\rho = 0.05$ and $\sigma^2 = 0.5$.

