# OpenReview forum: "Variational Learning Finds Flatter Solutions at the Edge of Stability"
_NeurIPS.cc/2025/Conference — NeurIPS 2025 spotlight_

### Official Review · Reviewer_BkQ3 · 2025-06-30

**Clarity:** 2
**Significance:** 3
**Originality:** 3
**Rating:** 4
**Confidence:** 3

**Summary:**

Variational Learning (VL) is reported to implicitly steer the optimization toward flatter minima by applying Gaussian perturbations to the weights, and this study theoretically analyzes this phenomenon from the perspective of the Edge of Stability. In particular, from the theoretical analysis of quadratic losses, the authors derive that the stability threshold of VL is smaller than that of Gradient Descent (GD), and they demonstrate that this threshold can be further lowered by enlarging the posterior variance and reducing the number of Monte Carlo samples. In addition, experiments on various neural networks confirm that the sharpness consistently adheres to the theoretically predicted Variational Factor, achieving lower sharpness and higher test accuracy than GD.

**Questions:**

See the weakness section. The following are additional questions.
- Although the paper interprets Theorem 1 as showing that VL is guided toward flatter minima because a stricter eigenvalue bound must be satisfied, Theorem 1 is only a sufficient condition for a single-step decrease in the loss. One could therefore equally conclude that VL is *less* stable, since the stricter bound makes a loss decrease harder to achieve. Could the authors discuss which interpretation is accurate and supply theoretical or empirical evidence to support that position?
- Hypothesis 1 assumes isotropic Gaussian noise, whereas the setting of Theorem 1 does not. Is this isotropic assumption intended to match the isotropic variance condition used in Theorem 3 to derive a necessary-and-sufficient bound, or is there another reason for the mismatch?
- Because the covariance matrix and \(Q\) are fixed, the algorithm keeps injecting noise from the same Gaussian distribution at every iteration, a situation that resembles Langevin dynamics or SGLD more than adaptive variational learning. If the work is viewed through a Langevin lens, how do the results connect with existing Langevin analyses, and what is genuinely novel for VL?
- A minor issue: after Appendix B the document unexpectedly reverts to Section 7 and Section 8, which is confusing; unifying the section numbering would improve readability.

**Ethical Concerns:**

["NO or VERY MINOR ethics concerns only"]

**Final Justification:**

All my concers are resolved

**Limitations:**

Yes

**Paper Formatting Concerns:**

No concern

**Quality:**

2

**Strengths And Weaknesses:**

## Strength
- The paper’s primary contribution is extending Edge-of-Stability sharpness analysis, which was previously limited to SGD variants, to the broadly used class of Gaussian-noise methods in variational learning.
- Through this extension, the authors rigorously show that the eigenvalue-based quadratic optimality condition for VL is stricter than that for SGD; this is a first-time result, and the accompanying numerical experiments confirm that this condition indeed governs optimization behavior.


## Weakness
- The theorem statements are mathematically slightly unclear; for example, Eq. (7) is defined after Theorem 1, whereas it should be defined within or before that theorem.
- Based on my reading of Appendix B (Sections 7 and 8 excluded), the mathematical link between Theorem 1 and Hypothesis 1 is unclear: Theorem 1 and Theorem 2 in Appendix merely give sufficient conditions for a one-step loss decrease under Gaussian-perturbed GD (and Theorem 3 adds necessity under special assumptions), yet Hypothesis 1 claims that the largest eigenvalue will hover around the derived threthold. So Theorem 1 is just the result for the one-step analysis, but the hypothesis considers the entire trajectory; this gap, central to the paper, requires a more rigorous discussion.
- Hypothesis 1 assumes isotropic Gaussian noise, whereas Theorem 1 does not; the paper does not explain this discrepancy.
- Because variational learning is characterized by adaptive variance, fixing both the noise covariance and the matrix \(Q\) makes the setup resemble Langevin dynamics or SGLD rather than genuine VL, raising doubts about the scope of the analysis.
- Practical VL employs minibatching, yet the paper offers no justification for ignoring minibatch-induced noise; if such noise is negligible, some justification should be provided.
- There is a substantial gap from Theorem 1 to the experimental sections: although Theorem 1 presumes fixed Gaussian noise, later experiments use Student-t noise and adaptive covariances that violate the theorem’s assumptions, leaving their relevance unclear.  As far as I understand, we need a significant modification to the proof of Theorem 1, which does not seem straightforward.
- Experiments are restricted to small 32 × 32 image datasets; evaluations on larger benchmarks such as ImageNet or NLP tasks are absent. I believe experiments for such ImageNet and NLP that follow the theoretical Gaussian setting would strengthen the paper more than the current Student-t or adaptive cases that diverge from the theory.

---

> ### Author Rebuttal · Authors · 2025-07-30
>
> We thank the reviewer for their time and effort in providing a critical and valuable feedback. We are happy to see the reviewer recognize the novelty and the strong empirical verification of the theorem. Below are our answer to the reviewer's comments to the weakness (W) and questions (Q).
>
> **W1**. Theorem statements are slightly unclear; for example, Eq. (7) is defined after Theorem 1, whereas it should be defined within or before that theorem.
>
> **A1**. Thanks for the suggestion. We will move Eq. 7 within the theorem.
>
> **W2**. Mathematical link between Theorem 1 and Hypothesis 1 is unclear.
>
> **A2** Similar gaps exist in existing EoS analysis for GD [1], SGD [2], and SAM [3]. All such works and ours rely on the principle that stability limits can be faithfully determined by local dynamics on a quadratic approximation. This is mentioned and discussed in lines 97-115. A rigorous proof is complex and beyond this paper’s scope, but extensions can be built on self-stabilization theory [4] which is a rather new development and not yet extended to settings such as ours. If it helps, we will add a dedicated appendix section to elaborate on this connection.
>
> **W3**: Hypothesis-1 assumes isotropic Gaussian noise, whereas Theorem-1 does not; the paper does not explain this discrepancy.
>
> **A3**: Thanks for pointing this out. This does not change the overall conclusion but has a technical implication. We will add an explanation in the paper. Essentially, the stability threshold is determined by the alignment between the noise covariance ($\mathbf{\Sigma}$) and the Hessian ($\mathbf{Q}$), a relationship captured by the term (Equation-23). For isotropic noise, this term becomes an equality $Tr(\mathbf{\Sigma}\mathbf{Q}^3)=\sigma^2 \sum_{i} \lambda^3_{i}$, but otherwise it forms an upper bound ($Tr(\mathbf{\Sigma}\mathbf{Q}^3) \leq \sum_{i} \sigma_{i}^2  \lambda^3_{i}$)   (lines 531-538). However, in both scenario, this provides a sufficient condition for expected descent so the difference is only in the technical details of the analysis.
>
> **W4 +Q3**: "the setup resemble Langevin dynamics or SGLD rather than genuine VL, raising doubts about the scope of the analysis. …. If the work is viewed through a Langevin lens, how do the results connect with existing Langevin analyses, and what is genuinely novel for VL?".
>
> **A4**:  There appears to be a misunderstanding. VL noise is added to the parameter which is fundamentally different from the SGLD noise added to the gradient. This is even true for a simple linear regression case, therefore it is not appropriate to view the method from a Langevin perspective. This difference is in fact highlighted in lines 203-214. We hope this clarifies the misunderstanding.
>
> **W5**: Practical VL employs minibatching, yet the paper offers no justification for ignoring minibatch-induced noise; if such noise is negligible, some justification should be provided.
>
> **A5**: Thanks for raising an important point. Our theoretical results intentionally focus on full gradients to separate the effect of posterior-sampling noise from the minibatch-gradient noise. The study of the interplay between the two is left to our experiments (Section 4.3) which confirms that minibatch noise further helps to reduce the stability threshold. A formal theory for the combined setting is challenging (due to heavy-tailed nature of the noise) and left as future work.
>
> **W6**: Theorem 1 presumes fixed Gaussian noise, later experiments use Student-t noise and adaptive covariances that violate the theorem’s assumptions, leaving their relevance unclear.
>
> **A6**: Thanks for pointing out this distinction. The later extensions are used to check if the theory extends to these tricky cases which are difficult to analyze. We find that the theory still holds even when the assumptions are violated. This only increases the relevance of the theoretical results showing that they are valid under much less restrictions. We will add some clarification in the text to help the readers understand the reason to use more complex posteriors. Our hope is that future work will extend the theory to other non-Gaussian cases.
>
> **W7**: NLP experiments that follow the theoretical Gaussian setting would strengthen the paper.
>
> **A7**: To verify that Variational GD with isotropic Gaussian noise achieves lower sharpness compared to GD on NLP tasks, we add a classification head to a frozen BERT-mini [5, 6] backbone and finetune the head on SST-2 [7], which is a sentiment classification dataset. For both GD and Variational GD, we use the same learning rate of 0.05.
> Here we show the results:
>
> **Variational GD with isotropic Gaussian noise**
> | Iteration | Train Loss | Val Loss | Sharpness |
> |-----------|------------|----------|-----------|
> | 10        | 0.6838     | 0.7149   | 29.2796   |
> | 20        | 0.6760     | 0.6715   | 30.7939   |
> | 50        | 0.6513     | 0.6803   | 29.9545   |
> | 100       | 0.6266     | 0.6598   | 34.0424   |
> | 200       | 0.6140     | 0.6159   | 33.9855   |
> | 500       | 0.5675     | 0.6119   | 35.6071   |
> | 1000      | 0.5657     | 0.5926   | 35.8911   |
> | 2000      | 0.5334     | 0.5806   | 34.4932   |
> | 5000      | 0.5232     | 0.5754   | 31.5243   |
> | 10000     | 0.5158     | 0.5644   | 31.7331   |
> | 20000     | 0.4924     | 0.5461   | 33.1990   |
> | 50000     | 0.4596     | 0.5320   | 32.1427   |
>
> **GD**
> | Iteration | Train Loss | Val Loss | Sharpness |
> |-----------|------------|----------|-----------|
> | 10        | 0.6834     | 0.6946   | 33.0345   |
> | 20        | 0.6735     | 0.6866   | 33.3173   |
> | 50        | 0.6502     | 0.6669   | 36.1241   |
> | 100       | 0.6183     | 0.6378   | 44.8439   |
> | 200       | 0.6192     | 0.6196   | 40.6111   |
> | 500       | 0.5766     | 0.5983   | 40.2430   |
> | 1000      | 0.5548     | 0.5882   | 40.1827   |
> | 2000      | 0.5414     | 0.5821   | 40.1149   |
> | 5000      | 0.5284     | 0.5741   | 40.0432   |
> | 10000     | 0.5138     | 0.5605   | 40.1765   |
> | 20000     | 0.4922     | 0.5477   | 40.3987   |
> | 50000     | 0.4570     | 0.5359   | 40.7298   |
>
>
> [1] Cohen, Jeremy M., et al. "Gradient descent on neural networks typically occurs at the edge of stability.", ICLR-21.
>
> [2] Lee, Sungyoon, and Cheongjae Jang. "A new characterization of the edge of stability based on a sharpness measure aware of batch gradient distribution." ICLR-23.
>
> [3] Long, Philip M., and Peter L. Bartlett. "Sharpness-aware minimization and the edge of stability." JMLR-24
>
> [4] Damian et al, "Self-Stabilization: The Implicit Bias of Gradient Descent at the Edge of Stability." ICLR-23.
>
> [5] Devlin, Jacob, et al. "BERT: Pre-training of Deep Bidirectional Transformers for Language Understanding." NAACL-19
>
> [6] Bhargava, Prajjwal, Aleksandr Drozd, and Anna Rogers. "Generalization in NLI: Ways (Not) To Go Beyond Simple Heuristics." arXiv
>
> [7] Socher, Richard, et al. "Recursive deep models for semantic compositionality over a sentiment treebank." EMNLP-13
>
>
> **Q1**. How does VL being less stable enables it to find flatter minima?
>
> **A**. You are correct that VL's stricter bound makes it less stable on a simple quadratic. However, in the complex landscapes of neural networks, this instability is precisely what guides it toward flatter solutions.
>
> Consider a landscape with two minima: one sharp and one flat. Gradient Descent (GD) with a small learning rate will simply converge to the nearest minimum, even if it's the undesirable sharp one. Variational Learning (VL), in contrast, has a stricter stability condition that is violated by the sharp minimum's high curvature. This instability prevents convergence there, effectively "kicking" the optimizer out of that sharp basin. It can then find and settle in the broad, flat minimum where the stability condition is satisfied. This process acts as an implicit filter, enabling VL to escape poor local minima and preferentially find the flatter solutions associated with better generalization.
>
> **Q2:** Use of isotropic assumption.
>
> **A**: You're correct, the use of isotropic noise in Hypothesis 1 is intentional. We use it to create a setting where our theoretical bound is tightest, allowing for a clear and direct empirical validation. Here’s the technical reasoning:
>
> *General (Sufficient) Bound*: Our main result (Theorem 3.1) provides a sufficient condition for descent that holds for any (diagonal) anisotropic noise. It's based on the trace inequality $Tr(\mathbf{\Sigma}\mathbf{Q}^3) \leq \sum_{i} \sigma_{i}^2  \lambda^3_{i}$, which is a conservative upper bound in the general case.
>
> *Isotropic Case*: For isotropic noise, the trace inequality becomes an exact equality. This makes the stability threshold upper bound tight, which is what we validate empirically under Hypothesis 1.
>
> Our condition is intentionally sufficient, as it's analytically tractable. It works by ensuring every term (line-541) in the expected loss change is negative. A necessary-and-sufficient condition (requiring the whole sum to be negative) is imposed on the whole Hessian eigen-spectrum, and other versions like Theorem 3 require strong assumptions (e.g., uniform alignment), making them less practical.
>
> We have revised the manuscript and Appendix E to clarify this distinction and to demonstrate why our sufficient condition is adequate for analyzing the neural network experiments.
>
> **Q3** is addressed with **W4**.
>
> **Q4** Numbering issue.
>
> **A**: Thank you for spotting the error in section number. We will fix it in our update.
>
> If the reviewer's concerns are addressed, we would be happy to see an increase in score.

---

> > ### Comment · Reviewer_BkQ3 · 2025-08-05
> > **Thank you for the reply**
> >
> > Thank you very much for your thorough and thoughtful responses. Most of my concerns have been addressed, and I will revise my score accordingly.
> >
> > I still have a couple of minor questions. These may not be the main focus of the paper, but I would appreciate your thoughts:
> >
> > - In scenarios where the Hessian spectrum is highly ill-conditioned or dominated by a few large eigenvalues, how reliable is the stability condition derived in Theorem 1? Could this potentially bias the analysis?
> >
> > - Although the experiments show that the theory appears to generalize beyond its assumptions, are there any specific examples or regimes where the theory breaks down (e.g., highly non-isotropic covariance)?

---

> ### Author Response · Authors · 2025-08-06
> **Thank you for your feedback and insightful questions**
>
> Thank you for the insightful questions. Here we answer the two questions.
>
> **Q1** *In scenarios where the Hessian spectrum is highly ill-conditioned or dominated by a few large eigenvalues, how reliable is the stability condition derived in Theorem 1? Could this potentially bias the analysis?*
>
> **A1**  *Short Answer:* Theorem-1 holds for a general quadratic Hessian spectrum (even when ill-conditioned). In this case, for stability to hold (expected loss, $\mathbb{E} [\Delta \ell]$ to decrease), the noise-variance in the (direction of the eigenvector with large eigenvalue) needs to be small which is given in condition (6) in paper. For stability to hold in direction of eigenvector with larger $\lambda_{i}$, $\sigma_{i}$ needs to be small due to monotonicity of VF(.). Below we give a detailed explanation.
>
> *Mathematical explanation:* The expected descent depends on the alignment of the noise covariance $\mathbf{\Sigma}$ and Hessian ($\mathbf{Q}$), through $Tr(\mathbf{\Sigma} \mathbf{Q}^3)$, that controls the stability, which is seen in the following equation:
>
> \begin{equation}
> \mathbb{E} [\Delta \ell] = - \rho \mathbf{g}^T \mathbf{g} + \frac{\rho^2}{2} \mathbf{g}^T \mathbf{Q} \mathbf{g}  + \frac{\rho^2}{2N_{s}}Tr(\mathbf{\Sigma} \mathbf{Q}^3)
> \end{equation}
>
> The value $Tr(\mathbf{\Sigma} \mathbf{Q}^3)$ is high when the noise is larger in the direction of eigenvector with large eigenvalue, hence maximizing $Tr(\mathbf{\Sigma} \mathbf{Q}^3)$ and making $\mathbb{E} [\Delta \ell]>0$. This is because maximum of $Tr(\mathbf{\Sigma} \mathbf{Q}^3)$  is $\sum_{i} \sigma_{i} \lambda^3_{i}$ (equality condition of Von Neuman's trace inequality) and is achieved when $\mathbf{\Sigma}$ and $\mathbf{Q}$ are aligned. Theorem-1 provides a sufficient condition by using this maximum alignment.
>
>
> **Q2** Although the experiments show that the theory appears to generalize beyond its assumptions, are there any specific examples or regimes where the theory breaks down (e.g., highly non-isotropic covariance)?
>
> **A2** Here are two cases we found where there is a slight mismatch between theory and experiment:
>
> 1) In experiments with vision transformers, the sharpness limit is usually observed to be slightly higher than the stability thresholds. This is not only the case with VL, but even for GD, the sharpness is higher than $\frac{2}{lr}$. This is a due to a phenomenon called attention entropy collapse commonly occurring in attention layers [1].
>
> 2) For non-isotropic noise, a small gap between our theoretical bound and empirical results can occasionally appear. This is because the theory is based on a worst-case analysis using the Von Neumann trace inequality $Tr(\mathbf{\Sigma} \mathbf{Q}^3) ≤ \sum_{i} \sigma^2_{i} \lambda^3_{i}$ , which considers maximal alignment between the noise and the Hessian. If the alignment during training is not exact, the bound is not perfectly tight, which can explain a slight mismatch. This differs from the isotropic case, where this bound is always an equality $Tr(\mathbf{\Sigma} \mathbf{Q}^3) =   \sigma^2 \sum_{i}\lambda^3_{i}$, leading to the perfect agreement between theory and experiments seen in our work.
>
> [1] Zhai, Shuangfei, et al. "Stabilizing transformer training by preventing attention entropy collapse." International Conference on Machine Learning. PMLR, 2023.
>
> We thank you again for your thorough review and time that helped improve this manuscript. We will add these detailed discussions in our revised version.

---

> > ### Comment · Area_Chair_CZ8B · 2025-08-08
> >
> > Dear Reviewer BkQ3,
> >
> > Thank you for your efforts in reviewing this paper. As the author-reviewer discussion period is drawing to a close, I wanted to kindly follow up and ask whether you have any thoughts on the authors' most recent response.
> >
> > All the best,
> > AC

---

> > ### Comment · Reviewer_BkQ3 · 2025-08-08
> >
> > Thank you for the clarification. Now all my concerns and questions are resolved. Best wishes

---

### Official Review · Reviewer_YUMo · 2025-07-01

**Clarity:** 3
**Significance:** 3
**Originality:** 3
**Rating:** 5
**Confidence:** 3

**Summary:**

The main contribution of the paper is the idea of using the concept of Edge of Stability (EoS) to analyze training dynamics and implicit regularization of variational inference for deep neural networks. After an introduction, the paper first recaps the basics of EoS for the quadractic loss, and then extends the analysis to variational inference in the simplified setting, where the variational family is the mean-field Gaussians with fixed and known variances.

Theorem 1 gives conditions for a decrease in the expected (quadratic) loss, and Lemma 2 gives conditions for a decrease in the actual loss (a high probability bouund). Using simple and well-designed set of numerical experiments, the paper provides insight into these results.

Next, the paper switches focus to larger models and proposes a hypothesis of how the top Hessian eigenvalue (of the loss) is related to the "stability threshold" (Hypothesis 1). Using several models (ViT, MLP, ResNet20) trained on CIFAR10, they provide evidence for this hypothesis, e.g. Fig3 suggests that  lower sharpness leads to better generalization and that the top Hessian eigenvalue tracks the stability threshold (Fig4). Moveover, they provide several sensitivity studies for different design choices (e.g. the posterior variance, the number of posterior samples, the effect of fat-tails in the variational family etc). In the last part of the paper, the authors relate the discussion to the Variational Online Newton (VON) rule and show empirically how different hyperparameters (batch size, learning rate) affect the optimization process.

**Questions:**

As mentioned above, I believe the paper could be further improved, if

- the authors would include a "global discussion" connecting the different observations and insights obtained through the paper and seeing into a bigger perspective.

- the authors would elaborate on whether these insights generalize to a more general variational setting, where the posterior covariance is not fixed. What would happen if the variational family was the mean-field Gaussians, low-rank or full-rank Gaussians?

- the authors would elaborate on the importance of the MSE loss function. To what degree do these findings extend to other losses?

Ideally, a summary of the related works should also be included in the main paper.




Minor details:
- Line 121: Typo "implmented"
- v_i is mentioned in Theorem 1, but not defined formally. (v_{1,t} is defined in section 4 and I assume v_i in Theorem has a similar definition)
- IVON abbrevation does not seems to be defined (Although the reference is included)
-

**Ethical Concerns:**

["NO or VERY MINOR ethics concerns only"]

**Final Justification:**

The authors addressed most of my concerns and, hence, I raised my score to 5.

**Quality:**

3

**Strengths And Weaknesses:**

Overall, the paper is well-written and relatively easy to follow. The idea of using EoS in the context of variational inference is very interesting and novel to the best of my knowledge. The paper provides novel theoretical results in a simplified setting, and then extends this to more complicated problems empirically. The theoretical analysis as well as the numerical experiments appears to be well-designed and sound.

In my opinion, the paper lacks a proper discussion. The final chapter "5 Conclusion and discussion" appears to me much more as a summary than a discussion. While the paper is well-written, the structure and coherence good be approved. To me, the paper appears a little bit as many small pieces (e.g. theoretical insights, small numerical experiments etc) that have been written independently. While most of sections contains a bit of discussion, it would be a significantly improvement (in my opinion) if the authors would include a "global" discussion in the end of the paper (there is indeed enough room), connecting the findings from each subsection and putting it into a bigger perspective. For example,

- The relation to PAC-Bayes. The abstract says: "Part of its empirical success can be explained by theories such as PAC-Bayes bounds....." and  the conclusion says: "Empirically, we observe that generalization in VL cannot be fully explained by PAC-Bayes bounds alone....", but besides that PAC-Bayes is not mentioned in the paper.

- The way I see paper is that the authors use the variational framework and approximate posterior as a means for regularization and stability, which is completely fine, but in many case the approximate posterior is also an object of interest in itself. Clearly, some of the results take advantage of the fact that the entropy is constant because the posterior covariance is fixed, but I believe the paper would benefit from a discussion of this and to what degree these results can be extended broader variational approximation, where the posterior covariance is not assumed to fixed.

-  The paper uses the MSE as a loss for the classification models. While I understand the choice in the context of the theoretical results, this choice is maybe not the most obvious choice in a general probabilistic setting, and  therefore, the paper would also benefit from a discussion of to what degrees these results would extend for example cross-entropy etc.

- Section 4.3 starts out by stating that: "One of the messages of this paper is that minimizing the variational objective (1) alone does not guarantee good generalization". I definitely agree with this statement, and I believe it is somewhat common knowledge in the field. Nevertheless, the section continues with an experiment, where it is shown that different optimization hyperparameter affects the optimization process and the final value of the variational objective. Yet, these results are not coupled in any to metrics quantifying generalization ability. Therefore, if the authors want to support their statement, they should indeed include some metric for generalization besides the variational objective itself.

---

> ### Author Rebuttal · Authors · 2025-07-31
>
> We thank the reviewer for their detailed and constructive feedback. We’re encouraged that the reviewer found the paper to be well-written, novel, and supported by strong experiments.
>
> We also thank the reviewer for suggesting to add a “global” discussion in the end of the paper. We are happy to revise it along the lines that reviewer suggested. We will do the following:
>
> 1) We will include a concrete discussion regarding PAC-Bayes, emphasizing the fact that PAC-Bayes focuses on generalization bounds [1] but don’t take an optimizer’s implicit regularization. Our work fills this gap by using  the Edge of Stability.
>
> 2) We will add a discussion on how the results can be extended to broader variational approximation, where the posterior covariance is not assumed to fixed (including low and full-rank Gaussians connecting them to Shampoo and MuON optimizers).
>
>       a) We will specifically highlight our numerical results using IVON (where covariance is learned) but also those with more flexible posterior (e.g., t-distribution). We will also highlight the difficulty of obtaining such theoretical results by using “preconditioned sharpness”. For VON, preconditioned sharpness is below 2/lr, while for Adam it’s around 2/lr [2]. As you noted, analyzing adaptive covariance is challenging because it requires jointly tracking the mean and preconditioner dynamics (Equation 8), even for a simple quadratic. We will highlight this finding and designate it as a direction for future work.
>
> 3) We will add a discussion on extending MSE to cross-entropy etc. We will also add some numerical results.
>
>       a) For, CE loss, the key difference lies in the peak sharpness. Due to its stricter stability bound, VL's sharpness peaks below  2/lr, whereas for GD, sharpness peaks to 2/lr. Following this peak, the sharpness for both methods decays to zero. Our preliminary results on a 2-layer MLP with CIFAR-10 confirm this: VL consistently exhibits lower sharpness than GD and decays faster. This decay to zero is a known property of the cross-entropy loss. The sharpness is governed by the Hessian, which in turn depends on the second derivative of the loss, $\ell^{''} \propto p_{i}(1-p_{i})$ where $p_{i}$ is the model's predicted probability. As training progresses and the model becomes confident, its predictions $p_{i}$ approach either 0 or 1. In both cases, the term $p_{i}(1-p_{i})$  approaches zero. This causes the entire Hessian, and thus the sharpness, to vanish. Here is an experiment with lr=0.05 trained with CE loss where the sharpness is reported across several epoch windows.
>
>
> | Iteration            |        GD       | VL (var=0.1) | VL (var=0.05) |
> |-------------------|--------------|---------------|-----------------|
> | 0-2000              |         45       |         26.       |         10           |
> | 2000-4000        |         40       |         22        |         10           |
> | End of training  |          8        |          3         |         1             |
>
>
> 4) We will add generalization metric to support the statement “One of the messages of this paper is that minimizing the variational objective (1) alone does not guarantee good generalization”.
>
>      a) We will clarify that what we aimed to say with that sentence is that we want to find good minimizers of the objective, and just blindly running a method without thinking about implicit bias is not enough. We will also add a sharpness plot in the figure alongside the variational objective to make it more clear and we report some of the metrics here alongside the original figure.
>
> | Iteration            |        BS=512     | BS=256   | BS=128 |
> |-------------------|--------------|---------------|-----------------|
> | 0-25k              |         37.6       |         22.3    |         15.4        |
> | 25k-40k          |         28.2       |         10.2    |         3.4      |
>
>
> [1] Alquier et al, “User-friendly introduction to PAC-Bayes bounds”.
>
>
> [2] Cohen et al, “Adaptive Gradient Methods at the Edge of Stability”. https://arxiv.org/abs/2207.14484
> Many thanks for pointing out typos. We will fix them!
>
> Many thanks for pointing out typos. We will fix them!

---

> > ### Comment · Reviewer_YUMo · 2025-08-05
> > **Thank you for the reply**
> >
> > Thank you for the response. It addresses most of my questions/concerns, and therefore, I have raised my score accordingly.

---

### Official Review · Reviewer_6LwC · 2025-07-03

**Clarity:** 4
**Significance:** 4
**Originality:** 4
**Rating:** 6
**Confidence:** 4

**Summary:**

The paper argues theoretically why introducing fixed weight noise during training with gradient descent (GD) leads to convergence to flatter regions of the loss function, verifies these predictions analytically in deep-learning models, and then extends both theory and empirical evaluation to variational learning (VL; i.e., where the noise amplitude is learned as well by maximizing the ELBO), and beyond GD to adaptive optimizes such as Adam. Theoretical predictions are *quantitative* and match empirical observations.

**Questions:**

In Figure 3, can you plot (in addition to the $2/\rho$ line) also a line that includes the "VF" factor (i.e., by solving the right part of Eq. 6 with equality instead of "<" numerically for $\lambda$)?

**Ethical Concerns:**

["NO or VERY MINOR ethics concerns only"]

**Final Justification:**

I've read the other reviews and didn't feel like I missed any substantial flaws, which is why I remain my rating of "strong accept".

As far as I understand, the main remaining criticism by another review are partially due to some misunderstandings of VL, and partially point out that the assumptions made for the theoretical derivations in the paper do not perfectly match the experimental setup. In my opinion, the latter is the only way how theory of deep learning can make any progress: without strong simplifying assumptions, I doubt that any nontrivial insight can be derived theoretically. But by clearly stating the assumptions made for theoretical derivations, we can identify which aspect of the (inevitably much more complicated) experimental setups can explain observed phenomena. Further, the paper does compare the theoretical findings to experiments, and I think that it convincingly shows that empirical findings do indeed match the derived theory.

**Limitations:**

Yes.

**Paper Formatting Concerns:**

Non noticed.

**Quality:**

4

**Strengths And Weaknesses:**

## Strengths

The writing is exceptionally clear. Edge of Stability theory is reviewed concisely but very clearly. Deriving the main result first in a simple setting (quadratic loss, Gaussian noise with fixed variance, GD with fixed learning rate and without preconditioning) makes it very clear where results are coming from and prepares readers for the later extensions.

*Qualitatively* the main result almost seems obvious in hindsight (the authors also explain it in simple terms without math on lines 52-53). But I see this as a strength rather than a weakness because (i) it speaks to the clarity of the presentation, (ii) both the theoretical predictions and the empirical evaluation go beyond qualitative statements and analyze the stability region *quantitatively*, and (iii) the quantitative analysis is eventually extended beyond the simple original case of fixed noise amplitude and learning rate to VL with an Adam optimizer.

## Minor weakness

I really can't find any weakness apart from a trivial point:

- When referring to Equations or Sections, only the number (without "Equation" or "Section") is rendered. This is somewhat confusing at first (e.g., on line 138: "An exact expression ... is derived in 6 where we show ..." --> I first checked Equation 6 before I realized that the authors are referring to *Section* 6)

---

> ### Author Rebuttal · Authors · 2025-07-31
>
> We thank the reviewer for their detailed feedback. We are glad that the reviewer found the writing to be *exceptionally clear*, results clear, and the experimental results perfectly validate the theory. Below is our response to some of your suggestions.
>
> **Q1**: Fixing the reference issue in numbering equations and sections.
>
> **A1**: Thanks for catching this. In the updated manuscript, we will properly refer to whether it is equation or section.
>
> **Q2**: Can you plot (in addition to the $2/\rho$ line) also a line that includes the "VF" factor?
>
> **A2**: Thanks for the suggestion! We will add it in the final version.

---

### Note · Authors · 2025-08-14

We thank all reviewers for their helpful feedback and positive evaluations, and we believe our rebuttal has addressed all questions. We are equally excited about this work, as it takes a first step towards understanding the mechanisms of Bayesian deep learning, linking its success to implicit regularization via optimization theory. We believe the work to be therefore of high interest to several communities of researchers at NeurIPS. Thank you again for your feedback which improved the manuscript.

---

### Decision · Program_Chairs · 2025-09-17

**Decision:**

Accept (spotlight)

**Comment:**

This paper analyses the training dynamics and implicit regularisation of variational inference for deep neural networks using Edge of Stability theory, showing that variational inference leads to flatter solutions than gradient descent. Reviewers found the paper exceptionally well written, with novel and sound theoretical contributions and well-designed experiments. While concerns were raised about theorem clarity, the connections between theory and experiments, and the scope of evaluation, the authors addressed these during rebuttal, and I recommend acceptance.